# Online Double Oracle

**Le Cong Dinh**[*]                                                     *l.c.dinh@soton.ac.uk*
*University of Southampton*

**Stephen McAleer**                                                      *smcaleer@uci.edu*
*University of California, Irvine*

**Zheng Tian**                                                *tianzheng@shanghaitech.edu.cn*
*ShanghaiTech University*

**Nicolas Perez-Nieves**                               *nicolas.perez14@imperial.ac.uk*
*Imperial College London*

**Oliver Slumbers**                                       *oliver.slumbers.19@ucl.ac.uk*
*University College London*

**David Henry Mguni**                                        *davidmguni@hotmail.com*
*Huawei R&D UK*

**Jun Wang**                                                             *w.j@huawei.com*
*Huawei R&D UK*

**Haitham Bou Ammar**                                      *haitham.ammar@huawei.com*
*Huawei R&D UK*

**Yaodong Yang**[†]                                       *yaodong.yang@pku.edu.cn*
*Institute for AI, Peking University*

**Reviewed on OpenReview:** *https://openreview.net/forum?id=rrMK6hYNSx*

## Abstract

Solving strategic games with huge action spaces is a critical yet under-explored topic in economics, operations research and artificial intelligence. This paper proposes new learning algorithms for solving two-player zero-sum normal-form games where the number of pure strategies is prohibitively large. Specifically, we combine no-regret analysis from online learning with Double Oracle (DO) from game theory. Our method—*Online Double Oracle (ODO)*—is provably convergent to a Nash equilibrium (NE). Most importantly, unlike normal DO, ODO is *rational* in the sense that each agent in ODO can exploit a strategic adversary with a regret bound of $\mathcal{O}(\sqrt{k \log(k)/T})$, where $k$ is not the total number of pure strategies, but rather the size of *effective strategy set*. In many applications, we empirically show that $k$ is linearly dependent on the support size of the NE. On tens of different real-world matrix games, ODO outperforms DO, PSRO, and no-regret algorithms such as Multiplicative Weights Update by a significant margin, both in terms of convergence rate to a NE, and average payoff against strategic adversaries [1].

## 1 Introduction

Understanding games with large action spaces is a critical topic in a variety of fields from economics to operations research and artificial intelligence. A key challenge is in computing a Nash equilibrium (NE) (Nash

---

[1][†]: Corresponding author. [*]: Part of this work was done as an intern at Huawei R&D UK, advised by Dr. Yaodong Yang.

Table 1: Properties of existing solvers on two-player zero-sum games $A_{n \times m}$. *:DO in the worst case has to solve all sub-games till reaching the full game, so the time complexity is one order magnitude larger than LP (van den Brand, 2020). †: Since PSRO uses approximate best-response, the total time complexity is unknown. ‡: Note that the regret bound of ODO can not be directly compared with the time complexity of DO, which are two different notions. ◇: Games with prohibitively large action spaces.

| Method | Rational (No-regret) | No Need to Know the Full Matrix $A$ | Time Complexity ($\tilde{\mathcal{O}}$) / Regret Bound ($\mathcal{O}$) | Large Games◇ |
|---|---|---|---|---|
| Linear Programming | | | $\tilde{\mathcal{O}}\big(n \exp(-T/n^{2.38})\big)$ | |
| Generalised Fictitious Play | | ✓ | $\tilde{\mathcal{O}}\big(T^{-1/(n+m-2)}\big)$ | |
| Multiplicative Weight Update | ✓ | ✓ | $\mathcal{O}\big(\sqrt{\log(n)/T}\big)$ | |
| Double Oracle | | ✓ | $\tilde{\mathcal{O}}\big(n \exp(-T/n^{3.38})\big)^{*}$ | ✓ |
| Policy Space Response Oracle | | ✓ | $\times^{\dagger}$ | ✓ |
| **Online Double Oracle** | ✓ | ✓ | $\mathcal{O}\big(\sqrt{k\log(k)/T}\big)^{\ddagger}$ | ✓ |

et al., 1950), where no player is better off by deviating from their current strategy. Unfortunately, finding a NE is generally intractable, and computing a two-player NE is known to be PPAD-hard (Chen & Deng, 2006). This holds true also for stochastic games (Deng et al., 2021). An exception is two-player zero-sum games where an NE can be tractably solved as a linear program (LP) (Morgenstern & Von Neumann, 1953). Despite the polynomial-time complexity of solving an LP (van den Brand, 2020), LP solvers are not adequate for games with prohibitively large action spaces. As a result, researchers have shifted their focus towards finding efficient approximation solutions (McMahan et al., 2003; Brown, 1951b; Zinkevich et al., 2007), or seek other solution concepts other than NE (Yang et al., 2020; 2018).

Double Oracle (DO) algorithm (McMahan et al., 2003) and its extension Policy Space Response Oracles (PSRO) (Lanctot et al., 2017) are efficient approaches to finding an approximate NE in games where the support of a NE is relatively small. In DO (McMahan et al., 2003), players are initialised with a subset of the full strategy space, thus playing only a sub-game of the original game; then, at each iteration, a best-response strategy to the NE of the last sub-game, which is assumed to be given by an *Oracle*, is added into each agent's strategy pool. The process stops when the best-response is already in the strategy pool or the performance improvement of the sub-game NE becomes trivial. When an exact best-response strategy is not available, an approximate solution is often adopted. For example, PSRO (Lanctot et al., 2017) applies reinforcement learning (RL) (Sutton & Barto, 2018) and multi-agent RL (Yang & Wang, 2020) oracles to approximate a best response.

Whilst DO and PSRO both provide an efficient way to approximate the NE in large-scale zero-sum games, they still face two open challenges. **Firstly**, they require both players to *coordinate* in order to solve the NE in sub-games and update the strategy set; both players have to follow the same learning dynamics such as Fictitious Play (FP) (Brown, 1951a) or implement LP to solve the sub-game NE together. This contradicts many real-world scenarios where an opponent can play any (non-stationary) strategy in sub-games.

**Secondly**, and more importantly, DO methods are not *rational* (Bowling & Veloso, 2001), in the sense that they do not provide a learning scheme that can exploit an adversary (i.e., achieving no-regret). Whilst a NE strategy guarantees the best performance in the worst scenario, it can be too pessimistic as a strategy compared to a rational strategy, sometimes lacking strategic diversity (Sanjaya et al., 2021; Yang et al., 2021). For example, in a repeated Rock-Paper-Scissors (RPS) game, playing the NE of $(1/3, 1/3, 1/3)$ every iteration makes one player un-exploitable. However, if the adversary acts irrationally and sticks to one strategy, say "Rock", then the player should exploit the adversary by consistently playing "Paper" to achieve larger rewards than playing the NE. No-regret algorithms (Cesa-Bianchi & Lugosi, 2006; Shalev-Shwartz et al., 2011) prescribe a learning scheme in which a player is guaranteed to achieve minimal regret against the best fixed strategy in hindsight when facing an unknown adversary (either rational or irrational). Notably, if both players follow no-regret algorithms, then it is guaranteed that their time-average policies will converge to a NE in zero-sum games (Blum & Monsour, 2007) . However, the regret bounds of popular no-regret

algorithms (Freund & Schapire, 1999; Auer et al., 2002) usually depend on the game size; for example, Multiplicative Weights Update (MWU) (Freund & Schapire, 1999) has a regret bound of $\mathcal{O}(\sqrt{\log(n)/T})$ and EXP3 (Auer et al., 2002) in the bandit setting has a regret of $\mathcal{O}(\sqrt{n\log(n)/T})$, where $n$ is the number of pure strategies (i.e., experts). As a result, directly applying no-regret algorithms, though rational, is not computationally feasible in solving large-scale games.

In this paper, we present a scalable solution to two-player zero-sum normal-form games where the game size (i.e., the number of pure strategies) is prohibitively large. Our main analytical tool is no-regret analysis from online learning (Shalev-Shwartz et al., 2011). Specifically, by conducting no-regret analysis (Freund & Schapire, 1999) within the DO framework (McMahan et al., 2003), we propose the *Online Double Oracle (ODO)* algorithm which inherits the key benefits from both sides. It is the first DO method that enjoys the no-regret property and can exploit unknown adversaries during game play. Importantly, ODO achieves a regret of $\mathcal{O}(\sqrt{k\log(k)/T})$ where $k$, the size of effective strategy set, is upper-bounded by the total number of pure strategies $n$ and often $k \ll n$ holds in practice. We test our algorithm on tens of games including random matrix games, real-world matrix games (Czarnecki et al., 2020), and Kuhn and Leduc Poker. Results show that in almost all games, ODO outperforms both DO and PSRO variants (Lanctot et al., 2017; McAleer et al., 2020), and the online learning baseline: MWU (Freund & Schapire, 1999) in terms of exploitability (i.e., distance to an NE) and average payoffs against different types of strategic adversaries.

## 2 Related Work

ODO contributes to both the game theory and online learning domains. To summarise our contribution, we present a list of existing game solvers for comparison in Table 1.

Approximating a NE has been extensively studied in the game theory literature. Fictitious Play (FP) (Brown, 1951b) and generalised FP (Leslie & Collins, 2006) are classical solutions where each player adopts a strategy that best responds to the time-average strategy of the opponent. Although FP is provably convergent to NE in zero-sum games, it is prohibited from solving large games due to the need to iterate through all pure strategies at each iteration; furthermore, the convergence rate depends exponentially on the game size (Brandt et al., 2010). In terms of large-scale zero-sum games, DO (McMahan et al., 2003; McAleer et al., 2021) and PSRO methods (Lanctot et al., 2017; McAleer et al., 2020; Feng et al., 2021; Perez-Nieves et al., 2021; Liu et al., 2021) have shown remarkable empirical success. For example, a distributed implementation of PSRO can handle games of size $10^{50}$ (McAleer et al., 2020). Yet, both FP and DO methods offer no knowledge about how to exploit an adversary in a game (i.e., no-regret property) (Hart & Mas-Colell, 2001), and thus are regarded as not *rational* (Bowling & Veloso, 2001). Modern solutions that are rational such as CFR methods (Zinkevich et al., 2007; Lanctot et al., 2009) are efficiently designed for extensive-form games only.

Algorithms with the no-(external) regret property can achieve guaranteed performance against the best-fixed strategy in hindsight (Shalev-Shwartz et al., 2011; Cesa-Bianchi & Lugosi, 2006), thus they are commonly applied to tackle adversarial environments. However, conventional no-regret algorithms such as Follow the Regularised Leader (Shalev-Shwartz et al., 2011), Multiplicative Weights Update (MWU) (Freund & Schapire, 1999) or EXP-3 (Auer et al., 2002) have regret bounds that are based on the number of pure strategies (i.e., experts). Moreover, these algorithms consider the full strategy set during updates, which hinders their applicability to large-scale games. In this paper, we leverage the advantages of both DO and no-regret learning to propose ODO. ODO enjoys the benefits of both being applicable to solving large games, and being able to maintain the no-regret property (i.e., being rational).

## 3 Notations & Preliminaries

A two-player zero-sum normal-form game is often described by a payoff matrix $\boldsymbol{A}$ of size $n \times m$. The rows and columns of $\boldsymbol{A}$ are the pure strategies of the row and the column players, respectively, and we consider $n$ and $m$ to be prohibitively large numbers. We denote the set of pure strategies for the row player as $\Pi := \{\boldsymbol{a}^1, \boldsymbol{a}^2, \dots \boldsymbol{a}^n\}$, and $C := \{\boldsymbol{c}^1, \boldsymbol{c}^2, \dots, \boldsymbol{c}^m\}$ for the column player.

The set of mixed strategies for the row-player is $\Delta_\Pi := \left\{ \boldsymbol{\pi} | \boldsymbol{\pi} = \sum_{i=1}^n x_i \boldsymbol{a}^i, \sum_{i=1}^n x_i = 1, x_i \geq 0, \forall i \in [n] \right\}$, and for the column player it is $\Delta_C := \{ \boldsymbol{c} | \boldsymbol{c} = \sum_{i=1}^m y_i \boldsymbol{c}^i, \sum_{i=1}^n y_i = 1, y_i \geq 0, \forall i \in [m] \}$. The support of a mixed strategy is written as $\text{supp}(\boldsymbol{\pi}) := \{ \boldsymbol{a}^i \in \Pi | x_i \neq 0 \}$, with its size being $|\text{supp}(\boldsymbol{\pi})|$.

We consider $\boldsymbol{A}_{i,j} \in [0, 1]$ to represent the (normalised) loss of the row player when playing a pure strategy $\boldsymbol{a}^i$ against the pure strategy $\boldsymbol{c}^j$ of the column player. At the $t$-th round, the expected payoff for the joint-strategy profile $(\boldsymbol{\pi}_t \in \Delta_\Pi, \boldsymbol{c}_t \in \Delta_C)$ is $(-\boldsymbol{\pi}_t^\top \boldsymbol{A} \boldsymbol{c}_t, \boldsymbol{\pi}_t^\top \boldsymbol{A} \boldsymbol{c}_t)$. In this paper, we consider the online setting in which players **do not** know the matrix $\boldsymbol{A}$, or the adversary's policy, but instead only receive a loss value after their strategy is played: e.g., at timestep $t + 1$, the row player observes $\boldsymbol{l}_t = \boldsymbol{A} \boldsymbol{c}_t$ from the environment and plays a new strategy $\boldsymbol{\pi}_{t+1}$. The goal of the players is to reach a Nash equilibrium.

**Nash Equilibrium.** A NE of a two-player zero-sum game can be defined by the minimax theorem (Neumann, 1928):

$$\min_{\boldsymbol{\pi} \in \Delta_\Pi} \max_{\boldsymbol{c} \in \Delta_C} \boldsymbol{\pi}^\top \boldsymbol{A} \boldsymbol{c} = \max_{\boldsymbol{c} \in \Delta_C} \min_{\boldsymbol{\pi} \in \Delta_\Pi} \boldsymbol{\pi}^\top \boldsymbol{A} \boldsymbol{c} = v, \tag{1}$$

for some $v \in \mathbb{R}$. The $(\boldsymbol{\pi}^*, \boldsymbol{c}^*)$ that satisfies Equation (1) is a NE of the game. In general, one can apply LP solvers to find the NE in small games (Morgenstern & Von Neumann, 1953). However, when $n$ and $m$ are large, the time complexity is not affordable. A more general solution concept is the $\epsilon$-Nash equilibrium.

**$\epsilon$-Nash Equilibrium.** For $\epsilon > 0$, we call a joint strategy $(\boldsymbol{\pi}, \boldsymbol{c}) \in \Delta_\Pi \times \Delta_C$ an $\epsilon$-NE if it satisfies

$$\max_{\boldsymbol{c} \in \Delta_C} \boldsymbol{\pi}^\top \boldsymbol{A} \boldsymbol{c} - \epsilon \leq \boldsymbol{\pi}^\top \boldsymbol{A} \boldsymbol{c} \leq \min_{\boldsymbol{\pi} \in \Delta_\Pi} \boldsymbol{\pi}^\top \boldsymbol{A} \boldsymbol{c} + \epsilon. \tag{2}$$

### 3.1 Double Oracle Method

The pseudocode for DO (McMahan et al., 2003) is listed in Appendix A.1. The DO method approximates a NE in large-scale zero-sum games by iteratively expanding and solving a series of sub-games (i.e., games with a restricted set of pure strategies). Since the sets of pure strategies of the sub-game are often much smaller than the original game, the NE of the sub-games can be easily solved via approaches such as FP. Based on the NE of the sub-game, each player finds a best-response to said NE, and expands their strategy set with this best-response. PSRO methods (Lanctot et al., 2017; McAleer et al., 2020) are a generalisation of DO in which RL methods (e.g., Actor Critic) are adopted to approximate the best-response strategy. In the worst case scenario (e.g., the support size of NE is large), DO may end up restoring the original game and will maintain no advantages over LP solutions.

Although DO can solve large-scale zero-sum games, it requires both players to *coordinate* by finding a NE in the sub-games; this is a problem for DO when applied in real-world scenarios, as it cannot exploit the opponent who can play any non-stationary strategy (see the example of RPS in the Introduction). ODO addresses this problem by combining DO with tools in online learning.

### 3.2 Online Learning

Solving for a NE in large-scale games is demanding, so an alternative approach is to consider learning-based methods. We believe that by playing the same game repeatedly, a learning algorithm can approximate the NE asymptotically. A common metric, *(external)-regret*, to quantify the performance of a learning algorithm is to compare its cumulated payoff with the best fixed strategy in hindsight.

**Definition 1** (No-Regret Algorithms). *Let $\boldsymbol{c}_1, \boldsymbol{c}_2, \ldots$ be a sequence of mixed strategies played by the column player, an algorithm of the row player that generates a sequence of mixed strategies $\boldsymbol{\pi}_1, \boldsymbol{\pi}_2, \ldots$ is called a no-regret algorithm if we have the following property hold.*

$$\lim_{T \to \infty} \frac{R_T}{T} = 0, \quad R_T = \max_{\boldsymbol{\pi} \in \Delta_\Pi} \sum_{t=1}^T \left( \boldsymbol{\pi}_t^\top \boldsymbol{A} \boldsymbol{c}_t - \boldsymbol{\pi}^\top \boldsymbol{A} \boldsymbol{c}_t \right).$$

If both players in a game follow a no-regret algorithm (not necessarily the same one), then the average strategies of both players converges to a NE (Cesa-Bianchi & Lugosi, 2006; Blum & Monsour, 2007). For

example, a well-known learning algorithm for games that has this no-regret property is the MWU algorithm (Freund & Schapire, 1999):

**Definition 2** (Multiplicative Weights Update). *Let $c_1, c_2, \ldots$ be a sequence of mixed strategies played by the column player. The row player is said to follow MWU if $\pi_{t+1}$ is updated as follows*

$$\pi_{t+1}(i) = \frac{\pi_t(i) \exp(-\mu_t {a^i}^\top A c_t)}{\sum_{i=1}^n \pi_t(i) \exp(-\mu_t {a^i}^\top A c_t)}, \ \forall i \in [n] \tag{3}$$

*where $\mu_t > 0$ is a parameter, $\pi_0 = [1/n, \ldots, 1/n]$ and $n$ is the number of pure strategies (a.k.a. experts).*

When $T$ is known in advance, by fixing the learning rate $\mu_t = \sqrt{8 \log(n)/T}$, we can achieve the optimal regret bound for MWU (Theorem 2.2 in (Cesa-Bianchi & Lugosi, 2006)):

$$\sqrt{T \log(n)/2}. \tag{4}$$

When $T$ is unknown, we can apply the Doubling Trick to achieve the regret bound of:

$$(\sqrt{2}/(\sqrt{2}-1))\sqrt{T \log(n)/2},$$

which is worse than the optimal one by a factor of $\sqrt{2}/(\sqrt{2}-1)$. Rooij et al. (De Rooij et al., 2014) proposed AdaHedge, a variant of MWU with adaptive learning rate $\mu_t = \log(n)/\Delta_{t-1}$ where $\Delta_t$ denotes the cumulative mixability gap [2]. Then following Theorem 8 in (De Rooij et al., 2014), the regret for AdaHedge will be bounded by:

$$\sqrt{T \log(n)} + 16/3 \log(n) + 2,$$

which is the worse than the optimal one by a factor of $\sqrt{2}$.

In our paper, w.l.o.g we use the optimal regret bound of MWU when $T$ is known to derive our theoretical results. In the case $T$ is unknown, following exactly the same argument with the Doubling Trick or AdHedge algorithm, we can derive similar regret bound up to a constant factor for our algorithms.

Note that since MWU requires updating the whole pure strategy set (of size $n$) at each iteration, it is not applicable to solving large-scale games.

## 4   Online Single Oracle

In this section, we introduce Online Single Oracle (OSO), a no-regret algorithm followed by individual players that can strategically exploit any non-stationary opponent unlike DO. Compared to the MWU algorithm, OSO can be applied to solving large zero-sum games as it only considers a smaller subset of the full pure strategy space. The following section is organised as follows: we start by setting out OSO, the key component of ODO. We then discuss the bound on the effective strategy set $k$, the key element in the regret bound of OSO. Finally, we set out two different questions on the effectiveness and efficiency of the best-response oracle, and analyse OSO's performance when the player only has access to *less-frequent* or *approximate* best-responses oracles.

### 4.1   Online Single Oracle Algorithm

One can think of OSO as an online counterpart to the *Single Oracle* in DO (McMahan et al., 2003) which can achieve the no-regret property. In contrast to classical no-regret algorithms such as MWU (Freund & Schapire, 1999) where the whole set of pure strategies needs considering at each iteration, i.e., Equation (3), we propose OSO that only considers a *subset* of the whole strategy set. The key operation is that, at each round $t$, OSO only considers adding a new strategy if it is the best-response to the average loss in a time window (defined formally in the following paragraph). As such, OSO can save on exploration costs by ignoring the pure strategies that have never been the best-response to any, so far observed, average losses, $\bar{l}$.

---

[2]See Appendix A.2 and A.3 for more details about Doubling Trick and AdaHedge algorithm.

---

**Algorithm 1:** Online Single Oracle Algorithm

---

1: **Input:** Player's pure strategy set $\Pi$
2: Init. effective strategies set: $\Pi_0 = \Pi_1 = \{\boldsymbol{a}^j\}, \boldsymbol{a}^j \in \Pi$
3: **for** $t = 1$ to T **do**
4:    **if** $\Pi_t = \Pi_{t-1}$ **then**
5:       Compute $\boldsymbol{\pi}_t$ by the MWU in Equation ([3](#))
6:    **else if** $\Pi_t \neq \Pi_{t-1}$ **then**
7:       Start a new time window $T_{i+1}$ and
       Reset $\boldsymbol{\pi}_t = \left[ 1/|\Pi_t|, \ldots, 1/|\Pi_t| \right], \;\; \bar{\boldsymbol{l}} = \boldsymbol{0}$
8:    **end if**
9:    Observe $\boldsymbol{l}_t$ and update the average loss in $T_i$: $\bar{\boldsymbol{l}} = \sum_{t \in T_i} \boldsymbol{l}_t / |T_i|$
10:    Calculate the best-response: $\boldsymbol{a}_t = \arg\min_{\boldsymbol{\pi} \in \Pi} \langle \boldsymbol{\pi}, \bar{\boldsymbol{l}} \rangle$
11:    Update the set of strategies: $\Pi_{t+1} = \Pi_t \cup \{\boldsymbol{a}_t\}$
12: **end for**
13: **Output:** $\boldsymbol{\pi}_T$, $\Pi_T$

---

Our OSO is listed in Algorithm [1](#). We initialise the OSO algorithm with a random strategy subset $\Pi_0$ from the original strategy set $\Pi$. Without loss of generality, we assume that $\Pi_0$ starts from only one pure strategy (line 2). We call subset $\Pi_t$ the **effective strategy set** at the timestep $t$, and define the period of consecutive iterations as one **time window** $T_i$ in which the effective strategy set stays fixed, i.e., $T_i := \{ t \mid |\Pi_t| = i \}$. At iteration $t$, we update $\boldsymbol{\pi}_t$ (line 5) whilst only considering the effective strategy set $\Pi_t$ (rather than whole set $\Pi$); and the best-response is computed against the average loss $\bar{\boldsymbol{l}}$ within the current time window $T_i$ (line 9). Adding a new best-response that is not in the existing effective strategy set will start a new time window (line 7). Notably, despite the design of effective strategy sets, the exact best-response oracle in line 10 still needs to search over the whole strategy set $\Pi$, which is a property that we relax through best-response approximation later.

We now present the regret bound of OSO as follows,

**Theorem 3** (Regret Bound of OSO). *Let $\boldsymbol{l}_1, \boldsymbol{l}_2, \ldots, \boldsymbol{l}_T$ be a sequence of loss vectors played by an adversary, and $\langle \cdot, \cdot \rangle$ be the dot product, OSO in Algorithm [1](#) is a no-regret algorithm with*

$$\frac{1}{T} \Big( \sum_{t=1}^{T} \langle \boldsymbol{\pi}_t, \boldsymbol{l}_t \rangle - \min_{\boldsymbol{\pi} \in \Pi} \sum_{t=1}^{T} \langle \boldsymbol{\pi}, \boldsymbol{l}_t \rangle \Big) \leq \frac{\sqrt{k \log(k)}}{\sqrt{2T}},$$

*where $k = |\Pi_T|$ is the size of the effective strategy set in the final time window.*

*Proof.* W.l.o.g, we assume the player uses the MWU as the no-regret algorithm and starts with only one pure strategy in $\Pi_0$ in Algorithm [1](#). Since in the final time window, the effective strategy set has k elements, there are exactly $k$ time windows. Denote $|T_1|, |T_2|, \ldots, |T_k|$ be the lengths of time windows during each of which the subset of strategies the no-regret algorithm considers does not change. In the case of finite set of strategies, $k$ will be finite and we have

$$\sum_{i=1}^{k} |T_k| = T.$$

In the time window with length $|T_i|$, following the regret bound of MWU in Definition [2](#) we have:

$$\sum_{t=|\bar{T}_i|+1}^{|\bar{T}_{i+1}|} \langle \boldsymbol{\pi}_t, \boldsymbol{l}_t \rangle - \min_{\boldsymbol{\pi} \in \Pi_{|\bar{T}_i|+1}} \sum_{t=|\bar{T}_i|+1}^{|\bar{T}_{i+1}|} \langle \boldsymbol{\pi}, \boldsymbol{l}_t \rangle \leq \sqrt{\frac{|T_i|}{2} \log(i)}, \quad \text{where } |\bar{T}_i| = \sum_{j=1}^{i-1} |T_j|. \tag{5}$$

In the time window $T_i$, we consider the full strategy set when we calculate the best response strategy in step 11 of Algorithm 1 and it stays in $\Pi_{|\bar{T}_i|+1}$. Therefore, the inequality (5) can be expressed as:

$$\sum_{t=|\bar{T}_i|+1}^{|\bar{T}_{i+1}|} \langle \boldsymbol{\pi}_t, \boldsymbol{l}_t \rangle - \min_{\boldsymbol{\pi} \in \Pi} \sum_{t=|\bar{T}_i|+1}^{|\bar{T}_{i+1}|} \langle \boldsymbol{\pi}, \boldsymbol{l}_t \rangle \leq \sqrt{\frac{|T_i|}{2} \log(i)}. \tag{6}$$

Sum up the inequality (6) for $i = 1, \dots k$ we have:

$$\sum_{i=1}^{k} \sqrt{\frac{|T_i|}{2} \log(i)} \geq \sum_{t=1}^{T} \langle \boldsymbol{\pi}_t, \boldsymbol{l}_t \rangle - \sum_{i=1}^{k} \min_{\boldsymbol{\pi} \in \Pi} \sum_{t=|\bar{T}_i|+1}^{|\bar{T}_{i+1}|} \langle \boldsymbol{\pi}, \boldsymbol{l}_t \rangle$$

$$\geq \sum_{t=1}^{T} \langle \boldsymbol{\pi}_t, \boldsymbol{l}_t \rangle - \min_{\boldsymbol{\pi} \in \Pi} \sum_{i=1}^{k} \sum_{t=|\bar{T}_i|+1}^{|\bar{T}_{i+1}|} \langle \boldsymbol{\pi}, \boldsymbol{l}_t \rangle = \sum_{t=1}^{T} \langle \boldsymbol{\pi}_t, \boldsymbol{l}_t \rangle - \min_{\boldsymbol{\pi} \in \Pi} \sum_{t=1}^{T} \langle \boldsymbol{\pi}, \boldsymbol{l}_t \rangle. \tag{7a}$$

Inequality (7a) is due to $\sum \min \leq \min \sum$. Using the Cauchy-Schwarz inequality we have:

$$\sum_{i=1}^{k} \sqrt{\frac{|T_i|}{2} \log(i)} \leq \sqrt{(\sum_{i=1}^{k} \frac{|T_i|}{2})(\sum_{i=1}^{k} log(i))} = \sqrt{\frac{T}{2}(\sum_{i=1}^{k} log(i))} \leq \sqrt{\frac{Tk \log(k)}{2}}.$$

Along with Inequality (7a) we can derive the regret:

$$\sqrt{\frac{Tk \log(k)}{2}} \geq \sum_{t=1}^{T} \langle \boldsymbol{\pi}_t, \boldsymbol{l}_t \rangle - \min_{\boldsymbol{\pi} \in \Pi} \sum_{t=1}^{T} \langle \boldsymbol{\pi}, \boldsymbol{l}_t \rangle.$$

$\square$

We note here that in line 7 of Algorithm 1, each time OSO enters a new time window, it sets equal weight for every pure strategy in the current effective strategy. Since we assume a fully adversarial environment, the historical data that the agent learnt in the previous time window does not provide any advantages over the current time window, thus in order to avoid any exploitation, the agent needs to reset the strategy as stated in Algorithm 1. In situations where priority knowledge can be observed through historical data, our OSO algorithm can exploit this knowledge by updating the starting strategy in each time window. We leave this important extension to our future work.

**Remark 1** (Worst-Case Regret Bound). Similar to all existing DO type of methods, in the worst-case scenario, OSO has to find all pure strategies, i.e., $k = |\Pi|$. Thus, the regret in the worst case scenario will be: $\sqrt{|\Pi| \log(|\Pi|)}/\sqrt{2T}$. However, we believe $k \ll |\Pi|$ holds in many practical cases such as against strategic adversary (Dinh et al., 2021a;b; Dinh, 2022). In later sections, we provide both theoretical and empirical evidence that real-world games tend to have $k \ll |\Pi|$.

In the next section, we discuss the relationship between the effective strategy set size $k$ and the full game size.

### 4.2 Size of Effective Strategy Set $k$

Heuristically, the practical success of ODO and other discussed methods (e.g., DO (McMahan et al., 2003) / PSRO (Lanctot et al., 2017)) is based on the assumption that the support size of the NE is small. Intuitively, since OSO is a no-regret algorithm, should the adversary itself follow a no-regret algorithm, the adversary's average strategy would converge to the NE. Thus, the learner's best-responses with respect to the average loss will include all the pure strategies in the support of the learner's NE. Therefore, under the assumption of DO and PSRO that the support size of NE is small, the effective strategy set size $k$ is potentially a far smaller number than the game size (i.e., $n$).

Note that the assumption of a NE having a small support size holds true in many situations. In **symmetric** games with random entries (i.e., see Theorem 2.8 in Jonasson et al. (2004)), it has been proved that the

expected support size of a NE will be $(\frac{1}{2} + \mathcal{O}(1))n$ where $n$ is the game size; showing that the support size of a NE strategy is only half of the game size. In **asymmetric** games with disproportionate action spaces (e.g., $n \gg m$), we provide the following lemma under which the support of a NE is small.

**Lemma 1.** *In asymmetric games $\boldsymbol{A}_{n \times m}, n \gg m$, if the NE $(\boldsymbol{\pi}^*, \boldsymbol{c}^*)$ is unique, then the support size of the NE will follow $|\operatorname{supp}(\boldsymbol{\pi}^*)| = |\operatorname{supp}(\boldsymbol{c}^*)| \leq m$.*

*(We provide the full proof in the Appendix C.1.)*

In situation where an asymmetric game $\boldsymbol{A}_{n \times m}, n \gg m$ does not has unique NE but it is nondegenerate [3], then following Proposition 3.3 in (Roughgarden, 2010), we can similarly bound the support of NE by $m$.

Empirically, we also show that the small support of a NE assumption holds on tens of real-world zero-sum games (Czarnecki et al., 2020) (see Table 2 in Appendix D) and randomly generated games (see Figure 1).

In the case when a dominant strategy exists, we can theoretically bound $k$ by the following lemma:

**Lemma 2.** *Suppose there exists a strictly dominant strategy for the player [4], then the size of the effective strategy set will be bounded by 2.*

As highlighted in Appendix C.4, OSO is theoretically more computationally efficient than MWU in convergence to NE when $k \leq \sqrt{n}$. Thus Lemma 1 with $m \leq \sqrt{n}$ and Lemma 2 provide subclasses of game in which OSO can theoretically outperform MWU.

Despite the practical success of our method and the DO/PSRO lines of work, there is no theoretical guarantee about the relationship between the support size of a NE and the performance of the algorithm. In this paper, we provide a negative result by constructing an example such that the size of the effective strategy set equals the size of the full strategy set, even when the support of NE is small.

**Lemma 3.** *Suppose the players start with the entry $\boldsymbol{A}_{1,1}$ and the game matrix $\boldsymbol{A}$ of the two-players zero-sum game is designed such as*

$$\boldsymbol{A}_{i,i} = 0.5 + \frac{0.1i}{n} \; \forall i \in [n]; \quad \boldsymbol{A}_{i,i+1} = 0.9 \; \forall i \in [n-1],$$

$$\boldsymbol{A}_{i,j} = 0.8 \; \forall j \geq i+2, i \in [n], \quad \boldsymbol{A}_{i,j} = \boldsymbol{A}_{i,i} + \frac{0.1}{2n} \forall j \leq i, i \in [n],$$

*where $n$ is the size of the pure strategy set for both players. Then the game has a unique Nash equilibrium with support size of 1 (i.e., the entry $\boldsymbol{A}_{n,n}$) and the effective strategy set in both DO and OSO will reach the size of pure strategy set, that is, $k = n$.*

We provide the full proof in Appendix C.3. The idea is that the the matrix $\boldsymbol{A}$ is designed such that the sub-game NE will change from $\boldsymbol{A}_{i,i}$ to $\boldsymbol{A}_{i,i+1}$ for $i \in [n]$, thus OSO will need to consider the full pure strategy set before reaching the game NE at $\boldsymbol{A}_{n,n}$. Following the same argument, when the players start with the entry $\boldsymbol{A}_{i,i}$, the effective strategy set will be $n - i + 1$ and thus when the players choose the starting entry as uniformly random, the expected size of effective stratetgy set will be: $\mathbb{E}(k) = (n+1)/2$. We would like to highlight that this negative result not only applies to our method, but also to **all** existing DO/PSRO algorithms and their variations.

However, as described in our experiments, we find that the extreme situation shown in Lemma 3 rarely occurs in practice. Later in Figure 1, we provide empirical evidence to support our claim that $k \ll |\Pi|$ and that there exists a linear relationship between $k$ and the Nash support size in many real-world applications.

### 4.3 OSO with Less-Frequent Best-Response

The first adaptation to the best-response process that we consider is to make calls to the best-response oracle less frequently. Obtaining a best-response strategy can be computationally expensive (Vinyals et al., 2019), and OSO considers adding a new best-response strategy at every iteration. A practical solution is to consider

---

[3]No mixed strategy of support size $h$ has more than h pure best responses
[4]we rigorously define in Appendix C.2.

adding a new strategy when the regret in the current time window exceeds a predefined threshold $\alpha$. To make OSO account for this, we denote $|\bar{T}_i| := \sum_{h=1}^{i-1} |T_h|$ as the starting point of the time window $T_i$, and write the threshold at $T_i$ as $\alpha_{t-|\bar{T}_i|}^i$ where $t - |\bar{T}_i|$ denotes the relative position of round $t$ in the time window $T_i$. We can make OSO add a new strategy only when the following condition is satisfied:

$$\min_{\boldsymbol{\pi} \in \Pi_t} \left\langle \boldsymbol{\pi}, \sum_{j=|\bar{T}_i|}^{t} \boldsymbol{l}_j \right\rangle - \min_{\boldsymbol{\pi} \in \Pi} \left\langle \boldsymbol{\pi}, \sum_{j=|\bar{T}_i|}^{t} \boldsymbol{l}_j \right\rangle \geq \alpha_{t-|\bar{T}_i|}^i. \tag{8}$$

Note that the larger the threshold $\alpha$, the longer OSO takes to add a new strategy into $\Pi_t$. However, choosing a large $\alpha$ will prevent the learner from acquiring the actual best-response, thus increasing the total regret $R_T$ by $\alpha$. In order to maintain the no-regret property, the $\alpha$ needs to satisfy

$$\lim_{T \to \infty} \frac{\sum_{i=1}^{k} \alpha_{T_i}^i}{T} = 0. \tag{9}$$

One choice of $\alpha$ that satisfies Equation (9) is $\alpha_{t-|\bar{T}_i|}^i = \sqrt{t - |\bar{T}_i|}$. We list the pseudo-code of OSO utilising Equation (8) and derive its regret bound of $\mathcal{O}\big(\sqrt{k \log(k)/T}\big)$ in Appendix B.1.

### 4.4 Considering $\epsilon$-Best Responses

The second adaptation brings OSO more closely in line with the work of PSRO by considering a non-exact best-response oracle. So far, OSO agents compute the exact best-response to the average loss function $\bar{l}$ (i.e., line 10 in Algorithm 1). Since calculating the exact best-response is often infeasible in large games, an alternative way is to consider an $\epsilon$-best response (e.g., through a RL subroutine similar to PSRO (Lanctot et al., 2017)) to the average loss.

By first analysing the convergence of DO, we can derive the regret bound as well as convergence guarantees for an OSO learner in the case of an $\epsilon$-best response oracle.

**Theorem 4.** *Suppose an OSO agent can only access the $\epsilon$-best response in each iteration when following Algorithm 1, if the adversary follows a no-regret algorithm, then the average strategy of the agent will converge to an $\epsilon$-NE. Furthermore, the algorithm is $\epsilon$-regret:*

$$\lim_{T \to \infty} \frac{R_T}{T} \leq \epsilon; \quad R_T = \max_{\boldsymbol{\pi} \in \Delta_\Pi} \sum_{t=1}^{T} \left( \boldsymbol{\pi}_t^\top \boldsymbol{A} \boldsymbol{c}_t - \boldsymbol{\pi}^\top \boldsymbol{A} \boldsymbol{c}_t \right).$$

*(We provide the full proof in Appendix B.2.)*

Theorem 4 justifies that in the case of approximate best-responses, OSO learners can still approximately converge to a NE. This results allows for the application of optimisation methods to approximate the best-response, which paves the way to use RL algorithm in solving complicated zero-sum games such as StarCraft (Vinyals et al., 2019; Peng et al., 2017). Now that we have introduced the algorithms the individual player will follow, we can now discuss the outcomes when both competing players follow such algorithms.

## 5 Online Double Oracle

Recall that if both players follow a no-regret algorithm, then the average strategies of both players converge to the NE in two-player zero-sum games (Cesa-Bianchi & Lugosi, 2006; Blum & Monsour, 2007). Since OSO has the no-regret property, it is then natural to study the self-play setting where both players utilise OSO, which, we call Online Double Oracle (ODO), and to investigate its convergence rate to a NE in large games.

There are two major benefits of using ODO in comparison to DO: **Firstly**, ODO does not need to compute a NE in each sub-game which can become computationally expensive for large sub-games. **Secondly**, ODO produces rational agents which can exploit an adversary to achieve the no-regret property. The convergence rate of ODO to a NE follows:

---

**Algorithm 2:** Online Double Oracle Algorithm

---

1: **Input:** Full pure strategy set $\Pi$, $C$
2: Init. effective strategies set: $\Pi_0 = \Pi_1, C_0 = C_1$
3: **for** $t = 1$ to T **do**
4:   Each player follows the OSO in Algorithm 1 with their respective effective strategy sets $\Pi_t, C_t$
5: **end for**
6: **Output**: $\boldsymbol{\pi}_T, \Pi_T, \boldsymbol{c}_T, C_T$

---

**Theorem 5.** *Suppose both players apply OSO. Let $k_1$, $k_2$ denote the size of the effective strategy set for each player. Then, the average strategies of both players converge to the NE with the rate:*

$$\epsilon_T = \sqrt{\frac{k_1 \log(k_1)}{2T}} + \sqrt{\frac{k_2 \log(k_2)}{2T}}.$$

*In the situation where both players follow OSO with a Less-Frequent Best-Response as in Equation (8) and $\alpha^i_{t-|\bar{T}_i|} = \sqrt{t - |\bar{T}_i|}$, the convergence rate to NE will be*

$$\epsilon_T = \sqrt{\frac{k_1 \log(k_1)}{2T}} + \sqrt{\frac{k_2 \log(k_2)}{2T}} + \frac{\sqrt{k_1} + \sqrt{k_2}}{\sqrt{T}}.$$

*Sketch Proof.* Using the regret bound of OSO we can derive that

$$\bar{\boldsymbol{\pi}}^\top \boldsymbol{A}\bar{\boldsymbol{c}} \geq \min_{\boldsymbol{\pi} \in \Pi} \boldsymbol{\pi}^\top \boldsymbol{A}\bar{\boldsymbol{c}} \geq \frac{1}{T}\sum_{t=1}^T \boldsymbol{\pi}_t^\top \boldsymbol{A}\boldsymbol{c}_t - \sqrt{\frac{k_1 \log(k_1)}{2T}} \geq \max_{\boldsymbol{c} \in C} \bar{\boldsymbol{\pi}}^\top \boldsymbol{A}\boldsymbol{c} - \sqrt{\frac{k_2 \log(k_2)}{2T}} - \sqrt{\frac{k_1 \log(k_1)}{2T}}.$$

Similarly, we have

$$\bar{\boldsymbol{\pi}}^\top \boldsymbol{A}\bar{\boldsymbol{c}} \leq \max_{\boldsymbol{c} \in C} \bar{\boldsymbol{\pi}}^\top \boldsymbol{A}\boldsymbol{c} \leq \frac{1}{T}\sum_{t=1}^T \boldsymbol{\pi}_t^\top \boldsymbol{A}\boldsymbol{c}_t + \sqrt{\frac{k_2 \log(k_2)}{2T}} \leq \min_{\boldsymbol{\pi} \in \Pi} \boldsymbol{\pi}^\top \boldsymbol{A}\bar{\boldsymbol{c}} + \sqrt{\frac{k_1 \log(k_1)}{2T}} + \sqrt{\frac{k_2 \log(k_2)}{2T}}.$$

Thus, with $\epsilon_T = \sqrt{\frac{k_1 \log(k_1)}{2T}} + \sqrt{\frac{k_2 \log(k_2)}{2T}}$ we have

$$\max_{\boldsymbol{c} \in C} \sum_{t=1}^T \bar{\boldsymbol{\pi}}^\top \boldsymbol{A}\boldsymbol{c} - \epsilon_T \leq \bar{\boldsymbol{\pi}}^\top \boldsymbol{A}\bar{\boldsymbol{c}} \leq \min_{\boldsymbol{\pi} \in \Pi} \boldsymbol{\pi}^\top \boldsymbol{A}\bar{\boldsymbol{c}} + \epsilon_T.$$

By definition, $(\bar{\boldsymbol{\pi}}, \bar{\boldsymbol{c}})$ is $\epsilon_T$-Nash equilibrium. Similarly, we can derive the convergence rate for OSO with Less-Frequent Best-Response. The full proof can be found in Appendix C.5. □

---

Theorem 5 suggests that, similar to OSO, the convergence rate of ODO will not depend on the game size, but rather the size of the effective strategy set of both players. As we studied in the OSO section, although we can not theoretically prove the linear relationship between the size of the effective strategy set and the size of the NE support, in many practical applications, such linear relationship holds true (shown in Figure 1). Thus, in the same way as DO, ODO will prove effective in solving large games where the size of the NE support is indeed much smaller than the full game size (e.g., also see Table 2 in Appendix D).

We note that the convergence of ODO and other baselines in our paper (i.e., MWU, CFR, XFP) are on average sense and recently many researchers have designed new algorithms that achieves last iterated convergence to NE, which is a stronger concept than average convergence (e.g., LRCA in (Dinh et al., 2021b), OMWU in (Daskalakis & Panageas, 2019)). However, these new algorithms inherit the computational complexity of conventional no-regret algorithms such as MWU and thus unable to apply in large-scale games.

### 5.1 ODO *vs.* Double Oracle with MWU

It is important to highlight that ODO is markedly different from simply implementing DO with MWU to solve for the sub-game NE (i.e., running MWU till convergence in the sub-game). Firstly, ODO checks for a best-response that is outside of the current effective strategy set per MWU update, whereas DO adds a best-response every time a sub-game NE is solved, which often requires thousands of MWU iterations. Most importantly, the best-response target in ODO (i.e., the time-average loss $\bar{l}$) is not necessarily a NE; this is in contrast to DO where the best-response is computed with respect to an exact NE. Intuitively, the convergence of DO only requires the best-response target in the last sub-game to be a NE, thus calculating the NEs in every sub-game to update the strategy set is not necessary. ODO tackles this problem by using the time-average loss as the new best-response target. As a result, ODO performs much better compared to DO in Figure 3. Finally, even if DO implements MWU to solve the sub-game NE, it is still not a no-regret algorithm. This also explains the performance gap between OSO and DO with MWU in Figure 4.

## 6 Experiments & Results

In this section, we aim to demonstrate the effectiveness of our practical use algorithms, OSO and ODO. However, we begin by looking at the major assumption of this work, and verify the linear dependence between the size of the effective strategy set and the NE support size in random matrix games. Next, over multiple real-world matrix games (Czarnecki et al., 2020), we evaluate OSO in both the self-play setting (i.e., ODO) and when playing against a strategic adversary. Finally, to validate performance on very large games, we show the performance of ODO on Kuhn and Leduc Poker which have huge size of pure strategies. As we benchmark on OSO iterations against other baselines, for fair comparison, we implement plain OSO in Algorithm 1 without the less-frequent best-response mentioned in Equation (8) for our experiments. All hyperparameter setting can be found in Appendix D.

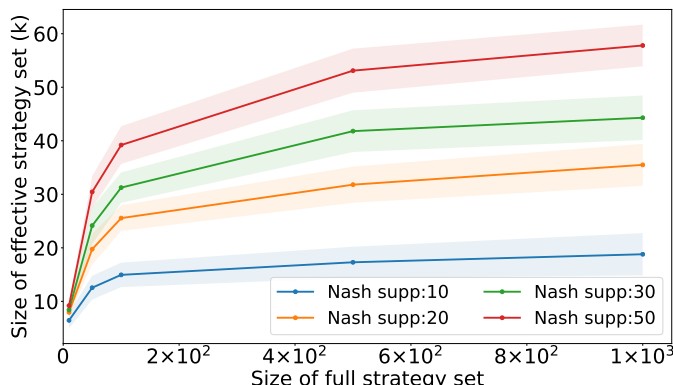

Figure 1: Sizes of effective strategy set (i.e., $k$) in cases of an OSO agent playing against an MWU opponent with different sizes of full strategy set and NE support. This plot shows that the size of OSO's effective strategy set does not increase drastically with the full strategy size, but rather depends on the support size of the NE.

### 6.1 Size of $k$ vs. Support Size of NE

We consider a set of zero-sum normal-form games of different sizes, the entries of which are sampled from a uniform distribution $\mathbf{U}(0, 1)$. We run OSO as the row player against a no-regret column player [5] until convergence, and plot the size of the OSO player's effective strategy set against its full strategy size. We run 20 seeds for each setting. As we can see from Figure 1, given a fixed support size of the NE, which is achieved by fixing the number of columns while increasing the number of rows in the game matrix, the size of the effective strategy set $k$ grows as the size of the full strategy set increases, but plateaus quickly. The larger the size of the NE support (not the full strategy set!), the higher this plateau will reach. Clearly, we can tell that the size of OSO's effective strategy set **does not** increase drastically with the full strategy size, but rather depends on the support size of the NE. This result confirms Theorem 3 in which we prove that OSO's regret bound depends on $k$, which is related to the size of the NE support but not the game size. Economically, this is a desired property as OSO can potentially avoid unnecessary computation, in contrast to other no-regret methods that require looping over the full strategy set at each iteration.

---

[5]We choose MWU for column player in our experiment, but we expect other no-regret algorithms would give a similar result.

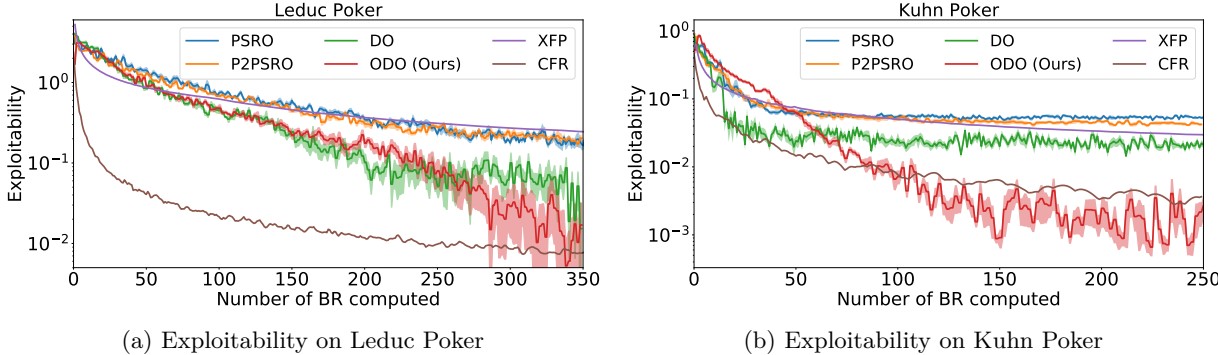

(a) Exploitability on Leduc Poker       (b) Exploitability on Kuhn Poker

Figure 2: Performance comparisons in exploitability on Poker games.

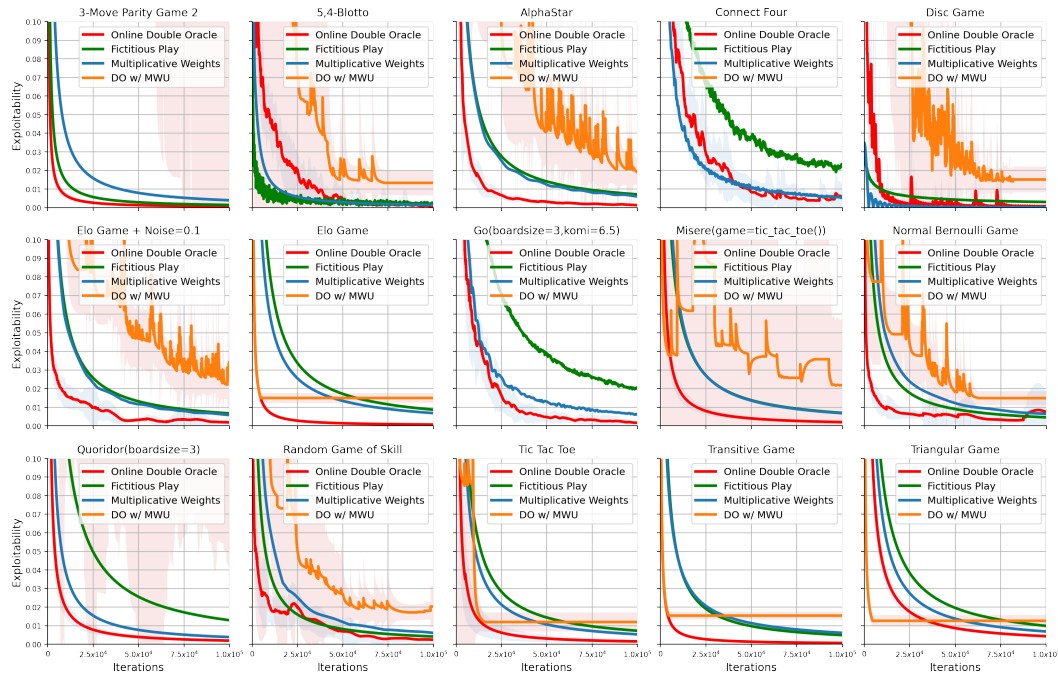

Figure 3: Performance comparisons under self-plays

## 6.2 Performance on Real-World Empirical Games

We investigate ODO in terms of convergence rate to a NE. To demonstrate its applicability to real-world problems, we replace the randomly generated normal-form games with 15 popular real-world zero-sum empirical games from Czarnecki et al. (2020). We compare the *exploitability* [6] (Davis et al., 2014) of ODO with other baseline methods (MWU, FP and DO [7]). We run each game with 20 seeds. In Figure 3, ODO outperforms the baselines in almost all 15 games. The advantage of ODO in terms of convergence rate over MWU and FP match our expectation as the support sizes of the NEs in these games are much smaller than the game sizes (reported in Table 2 in Appendix D). For DO with MWU, since it takes many iterations in each sub-game to converge to the NE, it performs poorly in comparison to ODO.

Apart from the self-play setting, we also look at the setting of playing against an MWU adversary in Figure 4. We can see that OSO outperforms MWU and DO baselines in average performance in almost all 15 games, which confirms the effectiveness of our design. Notably, MWU achieves a constant payoff; we believe this is because these games are symmetric and since both players follow MWU with the same learning rate, the

---

[6]exploitability= 0 implies the algorithm reaches the true NE.

[7]Following the discussion in the Online Double Oracle section and for fair comparisons, we implement DO by adopting MWU as the sub-game NE solver and report the total number of MWU iterations DO needs to achieve a low exploitability.

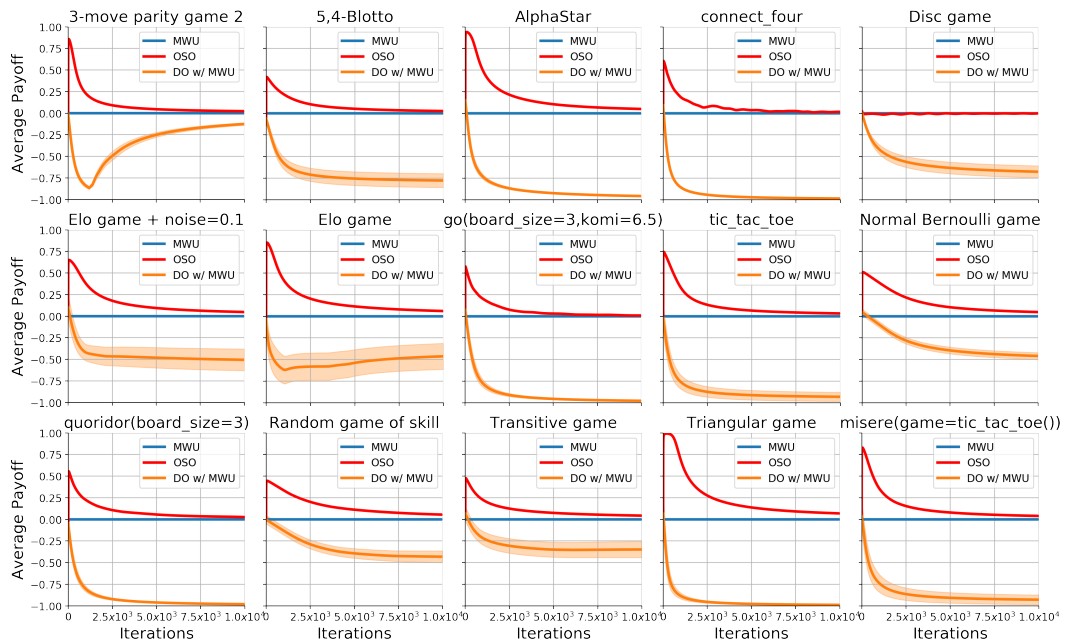

Figure 4: Performance comparisons against MWU adversary

payoff will always be the value of the game (thus the ground truth), which OSO will eventually converge to as well.

### 6.3 Performance on Poker Games

To further investigate ODO's effectiveness, we test ODO on Kuhn and Leduc Poker. Since ODO is designed only for normal-form games, we adopt the tabular setting (McAleer et al., 2020; Lanctot et al., 2017) in which an exact best-response is computed by a tree-traversal oracle (see OpenSpiel (Lanctot et al., 2019) or MALib (Zhou et al., 2021)), and for PSRO methods, we perturb the exact best-response with random noise. We benchmark how many times such a best-response oracle is called by different methods. We compare against the state-of-the-art PSRO method: P2SRO [8] (McAleer et al., 2020), and two extensive-form game solvers, CFR (Zinkevich et al., 2007) and XFP (Lanctot et al., 2019). As shown in Figure 2, ODO shows a significant improvement in exploitability compared to all existing DO and PSRO baselines, and it almost catches up with the CFR solver in Leduc Poker, and it outperforms CFR on Kuhn Poker. Importantly, ODO uses the fewest best-response calls to achieve the lowest exploitability. We believe these are promising results, since ODO is not designed for extensive-form games, but experimentally it matches up with efficient extensive-form methods. In our future work, we hope to use the idea of ODO and Regret Matching (Hart & Mas-Colell, 2000) to create a state-of-the-art solver in extensive-form games.

## 7 Conclusion

We propose a novel solver for two-player zero-sum games where the number of pure strategies $n$ is huge. Our method, *Online Double Oracle*, absorbs the benefits from both online learning methods and Double Oracle methods; it achieves the regret bound of $\mathcal{O}(\sqrt{k \log(k)/T})$ where $k$ is the size of the effective strategy set rather than the game size $n$. Importantly, ODO can exploit opponents during game play. In tens of real-world games, we show that ODO outperforms a series of algorithms including MWU, DO and PSRO both in terms of convergence rate to NE and average payoff against strategic adversaries.

---

[8]We have discounted the fact that P2SRO uses multiple workers (we use two) to compute best-responses.

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

# A    Appendix: Double Oracle and Variant of MWU

## A.1    Double Oracle

---

**Algorithm 3:** Double Oracle Algorithm (McMahan et al., 2003)

---

1: **Input:** Full strategy set $\Pi, C$
2: Initialise sets of strategies $\Pi_0, C_0$
3: **for** $t = 1$ to $\infty$ **do**
4:     **if** $\Pi_t \neq \Pi_{t-1}$ or $C_t \neq C_{t-1}$ **then**
5:         Solve the NE of the sub-game $\boldsymbol{G}_t$:
           $(\boldsymbol{\pi}_t^*, \boldsymbol{c}_t^*) = \arg\min_{\boldsymbol{\pi} \in \Delta_{\Pi_t}} \max_{\boldsymbol{c} \in \Delta_{C_t}} \boldsymbol{\pi}^\top \boldsymbol{A} \boldsymbol{c}$
6:         Find the best response $\boldsymbol{a}_{t+1}$ and $\boldsymbol{c}_{t+1}$ to $(\boldsymbol{\pi}_t^*, \boldsymbol{c}_t^*)$:
                 $\boldsymbol{a}_{t+1} = \arg\min_{\boldsymbol{a} \in \Pi} \boldsymbol{a}^\top \boldsymbol{A} \boldsymbol{c}_t^*$
                 $\boldsymbol{c}_{t+1} = \arg\max_{\boldsymbol{c} \in C} \boldsymbol{\pi}_t^{*\top} \boldsymbol{A} \boldsymbol{c}$
7:         $\Pi_{t+1} = \Pi_t \cup \{\boldsymbol{a}_{t+1}\}, C_{t+1} = C_t \cup \{\boldsymbol{c}_{t+1}\}$
8:     **else if** $\Pi_t = \Pi_{t-1}$ and $C_t = C_{t-1}$ **then**
9:         Terminate
10:    **end if**
11: **end for**

---

The pseudocode of DO algorithm (McMahan et al., 2003) is listed in Algorithm 3. The DO method approximates NE in large-size zero-sum games by iteratively creating and solving a series of sub-games (i.e., game with a restricted set of pure strategies). Specifically, at time step $t$, the DO learner solves the NE of a sub-game $\boldsymbol{G}_t$. Since the sets of pure strategies of the sub-game $\boldsymbol{G}_t = (\Pi_t, C_t)$ are often much smaller than the original game, the NE of sub-game $\boldsymbol{G}_t$ can be easily solved in line 5. Based on the NE of the sub-game: $(\boldsymbol{\pi}_t^*, \boldsymbol{c}_t^*)$, each player finds a best response to NE (line 6), and expand their strategy set (line 7). PSRO methods (Lanctot et al., 2017; McAleer et al., 2020) are variants of DO methods in which RL methods are adopted to approximate the best response strategy. For games where small support size assumption does not hold, DO methods restore to solve the original game and have no advantages over LP solutions.

Although DO can solve large-scale zero-sum games, it requires to *coordinate* both players to consistently find the NE (see line 5 in Algorithm 3); this renders an disadvantage of DO that when applied in real-world games, it cannot exploit the opponent who can play any non-stationary strategy (see the example of RPS in Introduction). Our ODO solution address this problem by combining DO with tools in online learning.

---

**Algorithm 4:** Online Double Oracle Algorithm

---

1: **Input:** Full pure strategy set $\Pi$, $C$
2: Init. effective strategies set: $\Pi_0 = \Pi_1, C_0 = C_1$
3: **for** $t = 1$ to T **do**
4:     Each player follows the OSO in Algorithm 1 with their respective effective strategy sets $\Pi_t, C_t$
5: **end for**
6: **Output**: $\boldsymbol{\pi}_T, \Pi_T, \boldsymbol{c}_T, C_T$

---

Recall that if both players follow a no-regret algorithm, then the average strategies of both players converge to the NE in two-player zero-sum games (Cesa-Bianchi & Lugosi, 2006; Blum & Monsour, 2007). Since OSO has the no-regret property, it is then natural to study the self-play setting where both players utilise OSO, which, we call Online Double Oracle (ODO), and to investigate its convergence rate to a NE in large games.

There are two major benefits of using ODO in comparison to DO: **Firstly**, ODO does not need to compute a NE in each sub-game which can become computationally expensive for large sub-games. **Secondly**, ODO produces rational agents which can exploit an adversary to achieve the no-regret property.

## A.2 Doubling Trick

**Definition 6** (**The Doubling Trick**). *For an algorithm with the regret $\alpha\sqrt{T}$ with a parameter depends on the time-horizon T. The Doubling Trick restarts the algorithm at round $2^m$ for m=0,1,2...*

Follow the doubling trick for an algorithm with regret $\alpha\sqrt{T}$, the total regret is not bigger than the sum of the regret in each part. We then have:

$$R_T \le \sum_{i=1}^{\lceil \log_2(T) \rceil} \alpha\sqrt{2^i} \le \alpha\frac{1-\sqrt{2T}}{1-\sqrt{2}} \le \frac{\sqrt{2}}{\sqrt{2}-1}\alpha\sqrt{T}.$$

## A.3 AdaHedge: MWU with Adaptive Learning Rate

**Definition 7** (AdaHedge (De Rooij et al., 2014)). *Let $\boldsymbol{c}_1, \boldsymbol{c}_2, ...$ be a sequence of mixed strategies played by the column player. The row player is said to follow AdaHedge if $\boldsymbol{\pi}_{t+1}$ is updated as follows*

$$\boldsymbol{\pi}_{t+1}(i) = \frac{\exp\left(-\mu_t \sum_{j=1}^{t-1} \boldsymbol{a}^{i\top}\boldsymbol{A}\boldsymbol{c}_j\right)}{\sum_{i=1}^{n} \exp\left(-\mu_t \sum_{j=1}^{t-1} \boldsymbol{a}^{i\top}\boldsymbol{A}\boldsymbol{c}_j\right)}, \; \forall i \in [n] \tag{10}$$

*where n is the number of pure strategies and $\mu_t > 0$ is an adaptive learning rate such as: $\mu_t = \log(n)/\Delta_{t-1}$. $\Delta_t$ denotes the cumulative mixability gap, which can be derived from historical data (Equation (5) in (De Rooij et al., 2014)).*

Then applying Theorem 8 in (De Rooij et al., 2014) to our setting with $S = 1$, $L^+ - L^- \le T$ we have:

$$\begin{aligned} R_T^{AdaHedge} &\le 2\sqrt{\frac{(L^+ - L^*)(L^* - L^-)}{L^+ - L^-}\log(n)} + \frac{16}{3}\log(n) + 2 \\ &\le 2\sqrt{\frac{(L^+ - L^-)^2/4}{L^+ - L^-}\log(n)} + \frac{16}{3}\log(n) + 2 \le \sqrt{T\log(n)} + \frac{16}{3}\log(n) + 2. \end{aligned} \tag{11}$$

# B Online Single Oracle

## B.1 OSO with Less-Frequent Best Response

**Theorem 8** (Regret Bound of OSO with Less-Frequent Best Response). *Let $\boldsymbol{l}_1, \boldsymbol{l}_2, \ldots, \boldsymbol{l}_T$ be a sequence of loss vectors played by an adversary. Then, OSO in Algorithm 5 is a no-regret algorithm with:*

$$\frac{1}{T}\Big( \sum_{t=1}^{T} \langle \boldsymbol{\pi}_t, \boldsymbol{l}_t \rangle - \min_{\boldsymbol{\pi} \in \Pi} \sum_{t=1}^{T} \langle \boldsymbol{\pi}, \boldsymbol{l}_t \rangle \Big) \le \frac{\sqrt{k\log(k)}}{\sqrt{2T}} + \frac{\sum_{i=1}^{k} \alpha_{|T_i|}^i}{T},$$

*where $k = |\Pi_T|$ is the size of effective strategy set in the final time window.*

*Proof.* W.l.o.g, we assume the player uses the MWU as the no-regret algorithm and starts with only one pure strategy in $\Pi_0$ in Algorithm 1. Since in the final time window, the effective strategy set has k elements, there are exactly $k$ time windows. Denote $|T_1|, |T_2|, \ldots, |T_k|$ be the lengths of time windows during each of which the subset of strategies the no-regret algorithm considers does not change. In the case of finite set of strategies, $k$ will be finite and we have

$$\sum_{i=1}^{k} |T_k| = T.$$

In the time window with length $|T_i|$, following the regret bound of MWU we have:

$$\sum_{t=|\bar{T}_i|+1}^{|\bar{T}_{i+1}|} \langle \boldsymbol{\pi}_t, \boldsymbol{l}_t \rangle - \min_{\boldsymbol{\pi} \in \Pi_{|\bar{T}_i|+1}} \sum_{t=|\bar{T}_i|+1}^{|\bar{T}_{i+1}|} \langle \boldsymbol{\pi}, \boldsymbol{l}_t \rangle \le \sqrt{\frac{|T_i|}{2}\log(i)}, \quad \text{where } |\bar{T}_i| = \sum_{j=1}^{i-1} |T_j|. \tag{12}$$

---

**Algorithm 5:** OSO with Less-Frequent Best Response

---

1: **Input:** A set $\Pi$ pure strategy set of player
2: $\Pi_0 := \Pi_1$: initial set of effective strategies;
3: **for** $t = 1$ to $\infty$ **do**
4:    **if** $\Pi_t = \Pi_{t-1}$ **then**
5:       Following the MWU update in Equation (3)
6:    **else if** $\Pi_t \neq \Pi_{t-1}$ **then**
7:       Start a new time window $T_{i+1}$
8:       Reset the MWU update in Equation (3) with a new initial strategy $\pi_t$
9:    **end if**
10:   Observe $l_t$ and update the average loss in the current time window $T_i$
        $\bar{l} = \frac{1}{|T_i|} \sum_{\pi_t \in T_i} l_t$ ;
11:   Calculate the best response:
        $a = \arg\min_{\pi \in \Pi} \langle \pi, \bar{l} \rangle$,
12:   **if** $\min_{\pi \in \Pi_{|\bar{T}_i|+1}} \langle \pi, \sum_{j=|\bar{T}_i|}^{t} l_j \rangle - \min_{\pi \in \Pi} \langle \pi, \sum_{j=|\bar{T}_i|}^{t} l_j \rangle \geq \alpha_{t-|\bar{T}_i|}^{i}$ **then**
13:      Update the strategy set: $\Pi_{t+1} = \Pi_t \cup a$
14:   **else**
15:      $\Pi_{t+1} = \Pi_t$
16:   **end if**
17:   Output the strategy $\pi_t$ at round $t$ for the player
18: **end for**

---

Since in the time window $T_i$, the size of the effective strategy set does not change, thus we have:

$$\min_{\pi \in \Pi_{|\bar{T}_i|+1}} \sum_{t=|\bar{T}_i|+1}^{|\bar{T}_{i+1}|} \langle \pi, l_t \rangle - \min_{\pi \in \Pi} \sum_{t=|\bar{T}_i|+1}^{|\bar{T}_{i+1}|} \langle \pi, l_t \rangle \leq \alpha_{|T_i|}^{i} \tag{13}$$

From Inequalities (12) and (13) we have:

$$\sum_{t=|\bar{T}_i|+1}^{|\bar{T}_{i+1}|} \langle \pi_t, l_t \rangle - \min_{\pi \in \Pi} \sum_{t=|\bar{T}_i|+1}^{|\bar{T}_{i+1}|} \langle \pi, l_t \rangle \leq \sqrt{\frac{|T_i|}{2} \log(i)} + \alpha_{|T_i|}^{i} \tag{14}$$

Sum up the inequality (14) for $i = 1, \ldots k$ we have:

$$\sum_{i=1}^{k} \left( \sqrt{\frac{|T_i|}{2} \log(i)} + \alpha_{|T_i|}^{i} \right) \geq \sum_{t=1}^{T} \langle \pi_t, l_t \rangle - \sum_{i=1}^{k} \min_{\pi \in \Pi} \sum_{t=|\bar{T}_i|+1}^{|\bar{T}_{i+1}|} \langle \pi, l_t \rangle$$

$$\geq \sum_{t=1}^{T} \langle \pi_t, l_t \rangle - \min_{\pi \in \Pi} \sum_{i=1}^{k} \sum_{t=|\bar{T}_i|+1}^{|\bar{T}_{i+1}|} \langle \pi, l_t \rangle = \sum_{t=1}^{T} \langle \pi_t, l_t \rangle - \min_{\pi \in \Pi} \sum_{i=1}^{T} \langle \pi, l_t \rangle \tag{15a}$$

$$\implies \sqrt{\frac{Tk \log(k)}{2}} + \sum_{i=1}^{k} \alpha_{|T_i|}^{i} \geq \sum_{t=1}^{T} \langle \pi_t, l_t \rangle - \min_{\pi \in \Pi} \sum_{i=1}^{T} \langle \pi, l_t \rangle. \tag{15b}$$

Inequality (15a) is due to $\sum \min \leq \min \sum$. Inequality (15b) comes from Cauchy-Schwarz inequality and Stirling' approximation. Thus, we have the derived regret bound. $\qquad\square$

## B.2 Proof of Theorem 4

Before provide the proof for the Theorem, we need the following lemma:

**Lemma 4.** *DO will converge to $\epsilon$-NE if players can only access to an $\epsilon$-best response in each round.*

*Proof.* We first prove in the case of single oracle algorithm. The double oracle proof will be similar. Since the number of strategies is finite, by the same argument in the the case of exact best response, the process will converge. Suppose that at time step $t$, the process stops. Since we use $\epsilon$-best response, we have the following relationship:

$$\boldsymbol{\pi_t}^\top A_t \boldsymbol{l_t} - \min_{\boldsymbol{\pi} \in \Pi} \boldsymbol{\pi}^\top \boldsymbol{A} \boldsymbol{l_t} \leq \epsilon$$

If we set the weight of pure strategies does not appear in $\boldsymbol{\pi_t}$ to be zero to make a $\boldsymbol{\pi'_t}$, then it is obvious that

$$\boldsymbol{\pi_t}^\top A_t = \boldsymbol{\pi'_t}^\top \boldsymbol{A}$$

Thus, we have the following relationship:

$$\boldsymbol{\pi'_t}^\top \boldsymbol{A} \boldsymbol{l_t} - \min_{\boldsymbol{\pi} \in \Pi} \boldsymbol{\pi}^\top \boldsymbol{A} \boldsymbol{l_t} \leq \epsilon \tag{16}$$

Further, since $\boldsymbol{l_t}$ is Nash equilibrium of $A_t$, we also have

$$\max_{l \in \Delta_l} \boldsymbol{\pi'_t}^\top A l - \boldsymbol{\pi'_t}^\top A l_t = \max_{l \in \Delta_l} \boldsymbol{\pi_t}^\top A_t l - \boldsymbol{\pi_t}^\top A_t \boldsymbol{l_t} = 0 \tag{17}$$

From inequalities (16) and (17), by definition we conclude that $(\boldsymbol{\pi'_t}, \boldsymbol{l_t})$ is $\epsilon$-NE of the game $\boldsymbol{A}$. $\square$

Now, we can prove Theorem 4:

**Theorem 9.** *Suppose OSO agent can only access the $\epsilon$-best response in each iteration when following Algorithm 1, if the adversary follows a no-regret algorithm, then the average strategy of the agent will converge to an $\epsilon$-NE. Furthermore, the algorithm is $\epsilon$-regret:*

$$\lim_{T \to \infty} \frac{R_T}{T} \leq \epsilon; \quad R_T = \max_{\boldsymbol{\pi} \in \Delta_\Pi} \sum_{t=1}^{T} \left( \boldsymbol{\pi_t}^\top \boldsymbol{A} \boldsymbol{c_t} - \boldsymbol{\pi}^\top \boldsymbol{A} \boldsymbol{c_t} \right).$$

*Proof.* Suppose that the player uses the Multiplicative Weights Update in Algorithm 1 with $\epsilon$-best response. Denote $|T_1|, |T_2|, \ldots, |T_k|$ be the lengths of time windows during each of which the subset of strategies the no-regret algorithm considers does not change. Furthermore,

$$\sum_{i=1}^{k} |T_k| = T.$$

In a time window $T_i$, the regret with respect to the best fixed strategy in the effective strategy set is:

$$\sum_{t=|\bar{T}_i|+1}^{|\bar{T}_{i+1}|} \langle \boldsymbol{\pi_t}, \boldsymbol{l_t} \rangle - \min_{\boldsymbol{\pi} \in \Pi_{|\bar{T}_i|+1}} \sum_{t=|\bar{T}_i|+1}^{|\bar{T}_{(i+1)}|} \langle \boldsymbol{\pi}, \boldsymbol{l_t} \rangle \leq \sqrt{\frac{|T_i|}{2} \log(i)}, \tag{18}$$

where $|\bar{T}_i| = \sum_{j=1}^{i-1} |T_j|$. Since in the time window $T_i$, the $\epsilon$-best response strategy stays in $\Pi_{\bar{T}_i+1}$ and therefore we have:

$$\min_{\boldsymbol{\pi} \in \Pi_{|\bar{T}_i|+1}} \sum_{t=|\bar{T}_i|+1}^{|\bar{T}_{(i+1)}|} \langle \boldsymbol{\pi}, \boldsymbol{l_t} \rangle - \min_{\boldsymbol{\pi} \in \Pi} \sum_{t=|\bar{T}_i|+1}^{|\bar{T}_{(i+1)}|} \langle \boldsymbol{\pi}, \boldsymbol{l_t} \rangle \leq \epsilon |T_i|$$

Then, from the inequality (18) we have:

$$\sum_{t=|\bar{T}_i|+1}^{|\bar{T}_{(i+1)}|} \langle \boldsymbol{\pi_t}, \boldsymbol{l_t} \rangle - \min_{\boldsymbol{\pi} \in \Pi} \sum_{t=|\bar{T}_i|+1}^{|\bar{T}_{(i+1)}|} \langle \boldsymbol{\pi}, \boldsymbol{l_t} \rangle \leq \sqrt{\frac{|T_i|}{2} \log(i)} + \epsilon |T_i|, \tag{19}$$

Sum up the inequality (19) for $i = 1, \ldots k$ we have:

$$\sum_{t=1}^{T} \langle \boldsymbol{\pi}_t, \boldsymbol{l}_t \rangle - \sum_{i=1}^{k} \min_{\boldsymbol{\pi} \in \Pi} \sum_{t=|\bar{T}_i|+1}^{|\bar{T}_{(i+1)}|} \langle \boldsymbol{\pi}, \boldsymbol{l}_t \rangle \leq \sum_{i=1}^{k} \sqrt{\frac{|T_i|}{2} \log(i)} + \epsilon |T_i|,$$

$$\Rightarrow \sum_{t=1}^{T} \langle \boldsymbol{\pi}_t, \boldsymbol{l}_t \rangle - \min_{\boldsymbol{\pi} \in \Pi} \sum_{i=1}^{k} \sum_{t=|\bar{T}_i|+1}^{|\bar{T}_{(i+1)}|} \langle \boldsymbol{\pi}, \boldsymbol{l}_t \rangle \leq \epsilon T + \sum_{i=1}^{k} \sqrt{\frac{|T_i|}{2} \log(i)} \tag{20a}$$

$$\implies \sum_{t=1}^{T} \langle \boldsymbol{\pi}_t, \boldsymbol{l}_t \rangle - \min_{\boldsymbol{\pi} \in \Pi} \sum_{t=1}^{T} \langle \boldsymbol{\pi}, \boldsymbol{l}_t \rangle \leq \epsilon T + \sum_{i=1}^{k} \sqrt{\frac{|T_i|}{2} \log(i)}$$

$$\implies \sum_{t=1}^{T} \langle \boldsymbol{\pi}_t, \boldsymbol{l}_t \rangle - \min_{\boldsymbol{\pi} \in \Pi} \sum_{t=1}^{T} \langle \boldsymbol{\pi}, \boldsymbol{l}_t \rangle \leq \epsilon T + \sqrt{\frac{T}{2}} \sqrt{k \log(k)}. \tag{20b}$$

Inequality (20a) is due to $\sum \min \leq \min \sum$. Inequality (20b) comes from Cauchy-Schwarz inequality and Stirling' approximation. Using inequality (20b), we have:

$$\min_{\boldsymbol{\pi} \in \Pi} \langle \boldsymbol{\pi}, \bar{l} \rangle \geq \frac{1}{T} \sum_{t=1}^{T} \langle \boldsymbol{\pi}_t, \boldsymbol{l}_t \rangle - \sqrt{\frac{k \log(k)}{2T}} - \epsilon. \tag{21}$$

That is, the OSO algorithm is $\epsilon$-regret in the case of $\epsilon$-best response.

Since the adversary follows a no-regret algorithm, we have:

$$\max_{l \in \Delta_L} \sum_{t=1}^{T} \langle \boldsymbol{\pi}_t, l \rangle - \sum_{t=1}^{T} \langle \boldsymbol{\pi}_t, \boldsymbol{l}_t \rangle \leq \sqrt{\frac{T}{2}} \sqrt{\log(L)}$$

$$\implies \max_{l \in \Delta_L} \langle \bar{\boldsymbol{\pi}}, l \rangle \leq \frac{1}{T} \sum_{t=1}^{T} \langle \boldsymbol{\pi}_t, \boldsymbol{l}_t \rangle + \sqrt{\frac{\log(L)}{2T}} \tag{22}$$

Using the inequalities in (21) and (22) we have:

$$\langle \bar{\boldsymbol{\pi}}, \bar{l} \rangle \geq \min_{\boldsymbol{\pi} \in \Pi} \langle \boldsymbol{\pi}, \bar{l} \rangle \geq \frac{1}{T} \sum_{t=1}^{T} \langle \boldsymbol{\pi}_t, \boldsymbol{l}_t \rangle - \sqrt{\frac{k \log(k)}{2T}} - \epsilon$$

$$\geq \max_{l \in \Delta_L} \langle \bar{\boldsymbol{\pi}}, l \rangle - \sqrt{\frac{\log(L)}{2T}} - \sqrt{\frac{k \log(k)}{2T}} - \epsilon$$

Similarly, we also have:

$$\langle \bar{\boldsymbol{\pi}}, \bar{l} \rangle \leq \max_{l \in \Delta_L} \langle \bar{\boldsymbol{\pi}}, l \rangle \leq \frac{1}{T} \sum_{t=1}^{T} \langle \boldsymbol{\pi}_t, \boldsymbol{l}_t \rangle + \sqrt{\frac{\log(L)}{2T}}$$

$$\leq \min_{\boldsymbol{\pi} \in \Pi} \langle \boldsymbol{\pi}, \bar{l} \rangle + \epsilon + \sqrt{\frac{k \log(k)}{2T}} + \sqrt{\frac{\log(L)}{2T}}$$

Take the limit $T \to \infty$, we then have:

$$\max_{l \in \Delta_L} \langle \bar{\boldsymbol{\pi}}, l \rangle - \epsilon \leq \langle \bar{\boldsymbol{\pi}}, \bar{l} \rangle \leq min_{\boldsymbol{\pi} \in \Pi} \langle \boldsymbol{\pi}, \bar{l} \rangle + \epsilon$$

Thus $(\bar{\boldsymbol{\pi}}, \bar{l})$ is the $\epsilon$-Nash equilibrium of the game. $\qquad \square$

## C  Bound on the size of effective strategy set $k$

### C.1  Proof of Lemma 1

**Lemma 5.** *In asymmetric games $\boldsymbol{A}_{n \times m}, n \gg m$, if the NE $(\boldsymbol{\pi}^*, \boldsymbol{c}^*)$ is unique, then the support size of the NE will follow $|\operatorname{supp}(\boldsymbol{\pi}^*)| = |\operatorname{supp}(\boldsymbol{c}^*)| \leq m$*

*Proof.* Since the size of $\boldsymbol{\pi}^*$ and $\boldsymbol{c}^*$ are $n$ and $m$ respectively, the size of the support of NE can not exceed the size of the game:

$$|\text{support}(\boldsymbol{\pi}^*)| \leq n; \ |\text{support}(\boldsymbol{c}^*)| \leq m.$$

In the case the game $\boldsymbol{A}$ has a unique Nash equilibrium, following Theorem 1 in (Bohnenblust et al., 1950), we have:

$$|\text{support}(\boldsymbol{\pi}^*)| = |\text{support}(\boldsymbol{c}^*)| \leq \min(n, m) = m.$$

Thus, we have proved the lemma. $\qquad\square$

## C.2  Proof of Lemma 2

**Definition 10** (Strictly Dominant Strategy)**.** *A strategy $\hat{\boldsymbol{\pi}}$ is called a strictly dominant strategy for the row player if:*

$$\hat{\boldsymbol{\pi}}^\top \boldsymbol{A}\boldsymbol{c} < \boldsymbol{\pi}^\top \boldsymbol{A}\boldsymbol{c} \ \ \forall \boldsymbol{\pi} \in \Pi, \boldsymbol{c} \in C.$$

**Lemma 6.** *Suppose there exists a strictly dominant strategy for the player, then the size of the effective strategy set will be bounded by 2.*

*Proof.* First we show that a strictly dominant strategy in two-player zero-sum game is a pure strategy. Let $\hat{\boldsymbol{\pi}}$ be the strictly dominant strategy. By definition of strictly dominant strategy we have:

$$\hat{\boldsymbol{\pi}}^\top \boldsymbol{A}\boldsymbol{c} < \boldsymbol{\pi}^\top \boldsymbol{A}\boldsymbol{c} \ \ \forall \boldsymbol{\pi} \in \Pi, \boldsymbol{c} \in C.$$

Let $\boldsymbol{a}^1$ be a pure strategy such that:

$$\boldsymbol{a}^1 = \underset{\boldsymbol{a} \in \Pi}{\text{argmin}}\, \boldsymbol{a}^\top \boldsymbol{A}\boldsymbol{c}^1,$$

where $\boldsymbol{c}^1$ is a constant vector. If $\hat{\boldsymbol{\pi}}$ is a mixed strategy then we have $\hat{\boldsymbol{\pi}} \neq \boldsymbol{a}^1$ and

$$\hat{\boldsymbol{\pi}}^\top \boldsymbol{A}\boldsymbol{c}^1 \geq {\boldsymbol{a}^1}^\top \boldsymbol{A}\boldsymbol{c}^1,$$

contradicts with the definition of strictly dominant strategy. Thus, the strictly dominant strategy in two-player zero-sum game is a pure strategy [9].

Now, after the first iteration, the OSO algorithm will add the best response to the effective strategy set. Since there exists a strictly dominant strategy $\hat{\boldsymbol{\pi}}$ and it is a pure strategy, $\hat{\boldsymbol{\pi}}$ will be added to the effective strategy set. From the second iteration, since the strictly dominant strategy $\hat{\boldsymbol{\pi}}$ is already in the effective strategy set, the best response to any average loss is always in the effective strategy set. Thus, the effective strategy set will not be expanded after iteration 2. In other words, the size of the effective strategy set will be bounded by 2. $\qquad\square$

## C.3  Proof of Lemma 3

**Lemma 7.** *Suppose the players start with the entry $\boldsymbol{A}_{1,1}$ and the game matrix $\boldsymbol{A}$ of the two-players zero-sum game is designed such as*

$$\boldsymbol{A}_{i,i} = 0.5 + \frac{0.1i}{n}\ \forall i \in [n]; \ \ \boldsymbol{A}_{i,i+1} = 0.9\ \forall i \in [n-1],$$

$$\boldsymbol{A}_{i,j} = 0.8\ \forall j \geq i+2, i \in [n],$$

$$\boldsymbol{A}_{i,j} = \boldsymbol{A}_{i,i} + \frac{0.1}{2n}\forall j \leq i, i \in [n],$$

*where $n$ is the size of the pure strategy set for both players. Then the game has a unique Nash equilibrium with support size of 1 (i.e., the entry $\boldsymbol{A}_{n,n}$) and the effective strategy set both DO and OSO method will reach the size of pure strategy set: $k = n$.*

---

[9]With the same argument, a strictly dominant strategy in any normal-form game is a pure strategy.

*Proof.* First, we show that the game $\boldsymbol{A}$ has a unique Nash equilibrium at entry $\boldsymbol{A}_{n,n}$. Following the design of matrix $\boldsymbol{A}$ we have:

$$\boldsymbol{a}^n = \operatorname*{argmin}_{\boldsymbol{a}\in\Pi} \boldsymbol{a}^\top \boldsymbol{A}\boldsymbol{c}^n; \;\; \boldsymbol{c}^n = \operatorname*{argmax}_{\boldsymbol{c}\in C} \boldsymbol{a}^{n\top} \boldsymbol{A}\boldsymbol{c}$$

Thus by definition, $(\boldsymbol{a}^n, \boldsymbol{c}^n)$ is the NE of the game and $\boldsymbol{A}_{n,n}$ is the minimax value $v$ of the game. Suppose there is another NE of the game $(\hat{\boldsymbol{a}}, \hat{\boldsymbol{c}})$ such that $\hat{\boldsymbol{a}} \neq \boldsymbol{a}^n$ [10]. Then, by definition of the NE we have:

$$\hat{\boldsymbol{a}}^\top \boldsymbol{A}\boldsymbol{c}^n \leq \hat{\boldsymbol{a}}^\top \boldsymbol{A}\hat{\boldsymbol{c}} = v = \boldsymbol{A}_{n,n},$$

since the minimax value $v$ is unique in two-player zero-sum game. However, by the design of matrix $\boldsymbol{A}$, $\boldsymbol{a}^\top \boldsymbol{A}\boldsymbol{c}^n \geq v \;\forall \boldsymbol{a} \in \Pi$ and the equal sign holds true only if $\boldsymbol{a} = \boldsymbol{a}^n$. This leads to a contradiction. Thus, the game $\boldsymbol{A}$ has a unique pure NE with respect to the entry $\boldsymbol{A}_{n,n}$.

Next, we show that Double Oracle method with the NE as best response target will recover the whole pure strategy set in this game.

By definition, the game start with the entry $\boldsymbol{A}_{1,1}$ and the initial effective strategy sets are $\Pi_0 = \{\boldsymbol{a}^1\}$ and $C_0 = \{\boldsymbol{c}^1\}$. Since, $\boldsymbol{A}_{1,1} = \operatorname{argmin}_{i\in[n]} \boldsymbol{A}_{i,1}$ and $\boldsymbol{A}_{1,2} = \operatorname{argmax}_{j\in[n]} \boldsymbol{A}_{1,j}$, the new sub-game 2 is created with the corresponding effective strategy set: $\Pi_1 = \{\boldsymbol{a}^1\}$ and $C_1 = \{\boldsymbol{c}^1, \boldsymbol{c}^2\}$. Note that the effective strategy set of the row player remains unchanged in this iteration. The NE in the sub-game 2 is the with respect to entry $\boldsymbol{A}_{1,2}$, thus in the next iteration, the best response targets for the column and row player are with respect to $\boldsymbol{a}^1$ and $\boldsymbol{c}^2$, respectively. Now, in iteration 3, since $\boldsymbol{A}_{2,2} = \operatorname{argmin}_{i\in[n]} \boldsymbol{A}_{i,2}$ and $\boldsymbol{A}_{1,2} = \operatorname{argmax}_{j\in[n]} \boldsymbol{A}_{1,j}$, the new sub-game 3 is created with the corresponding effective strategy set: $\Pi_2 = \{\boldsymbol{a}^1, \boldsymbol{a}^2\}$ and $C_1 = \{\boldsymbol{c}^1, \boldsymbol{c}^2\}$. Note that in this round, the effective strategy set of the column player remains unchanged. Following the same process, the effective strategy set of the row player will add the $\boldsymbol{a}^i$ pure strategy in iteration $2i-1$ while the effective strategy set of the column player will add the pure strategy $\boldsymbol{c}^i$ at iteration $2i-2$. Therefore, the DO method will add the whole pure strategy set until converging to the NE in this example.

For the OSO method, we can follow the same process in the above DO case. That is, for the adversary, we allow it to play the NE of the sub-game in the same order as in DO. Since OSO is a no-regret algorithm and the average loss will remain the same for each time window (since we fix the adversary in this case), the OSO algorithm will converge to the best response with respect to the current average loss. Since we design the game such that in each sub-game, the Nash Equilibrium will be a pure strategy, the best response with respect to the NE of the adversary will also be the NE of the player in the current sub-game. After the player (i.e., following OSO method) converges to the NE of the sub-game, the adversary will move to play the NE of the next sub-game. That way, the OSO algorithm will need to add the whole pure strategy set when playing against this type of adversary.

In a more specific way, the adversary can play the following policy. In the first iteration, the adversary plays $\boldsymbol{c}^1$. Since the best reponse with respect to $\boldsymbol{c}^1$ is $\boldsymbol{a}^1$, $\boldsymbol{a}^1$ will be added to the effective strategy set. Then at iteration 2, the adversary plays $\boldsymbol{c}^2$. Then, $\boldsymbol{a}^2$ is the best response with respect to the current average loss (i.e., $\boldsymbol{A}\boldsymbol{c}^1 + \boldsymbol{A}\boldsymbol{c}^2$). Thus, $\boldsymbol{a}^2$ is added to the effective strategy set. In the next iteration, by the design of the game matrix $\boldsymbol{A}$, we have the following relationship:

$$\boldsymbol{a}^{i+h} = \operatorname*{argmin}_{\boldsymbol{a}\in\Pi} \boldsymbol{a}^\top \boldsymbol{A}(\sum_{j=i}^{i+h} \boldsymbol{c}^j) \;\forall i, h > 0.$$

Thus, by letting the adversary plays $\boldsymbol{c}^1$ to $\boldsymbol{c}^n$ sequentially, the effective strategy set of the agent will need to recover the whole pure strategy set. Note that the policy used by the adversary is also a no-regret algorithm since in the later rounds, the adversary can just follows the pure strategy $\boldsymbol{c}^n$ and achieving the value $v$ of the game, thus deducting any negative payoffs from the first $n$ rounds. $\qquad\square$

---

[10]The same argument holds true in the case $\boldsymbol{c}^n \neq \hat{\boldsymbol{c}}$.

### C.4 Computation Comparison between ODO and MWU

**Lemma 8.** *when the size of effective strategy set $k$ satisfies*

$$k \le \sqrt{n},$$

*then ODO is more computationally efficient than MWU in finding Nash Equilibrium in two-player zero-sum games.*

*Proof.* In order to find $\epsilon$-NE, ODO requires

$$\sqrt{\frac{k \log(k)}{T}} \le \epsilon \iff T \ge \frac{k \log(k)}{\epsilon^2},$$

where $T$ is the number of rounds. Since each round requires $O(k)$ computation, the total computation of ODO will be $\frac{k^2 \log(k)}{\epsilon^2}$.

Similarly, to find $\epsilon$-NE, MWU requires $\sqrt{\log(n)/T} \le \epsilon$ and computation for each round is $O(n)$. Therefore, the overall computation of MWU will be approximately:

$$\frac{n \log(n)}{\epsilon^2}.$$

Therefore, ODO is more computationally efficient than MWU when

$$\frac{k^2 \log(k)}{\epsilon^2} \le \frac{n \log(n)}{\epsilon^2},$$

which can be satisfied when $k \le n$. $\qquad\qquad\square$

### C.5 Convergence Rate of ODO

**Theorem 11.** *Suppose both players apply OSO. Let $k_1$, $k_2$ denote the size of the effective strategy set for each player. Then, the average strategies of both players converge to the NE with the rate:*

$$\epsilon_T = \sqrt{\frac{k_1 \log(k_1)}{2T}} + \sqrt{\frac{k_2 \log(k_2)}{2T}}.$$

*In the situation where both players follow OSO with a Less-Frequent Best-Response as in Equation (8) and $\alpha^i_{t-|\bar{T}_i|} = \sqrt{t - |\bar{T}_i|}$, the convergence rate to NE will be*

$$\epsilon_T = \sqrt{\frac{k_1 \log(k_1)}{2T}} + \sqrt{\frac{k_2 \log(k_2)}{2T}} + \frac{\sqrt{k_1} + \sqrt{k_2}}{\sqrt{T}}.$$

*Proof.* Using the regret bound of OSO algorithm in Theorem 3 we have:

$$\sum_{t=1}^{T} \boldsymbol{\pi}_t^\top \boldsymbol{A} \boldsymbol{c}_t - \min_{\boldsymbol{\pi} \in \Pi} \sum_{t=1}^{T} \boldsymbol{\pi}^\top \boldsymbol{A} \boldsymbol{c}_t \le \sqrt{\frac{T k_1 \log(k_1)}{2}}; \quad \max_{\boldsymbol{c} \in C} \sum_{t=1}^{T} \boldsymbol{\pi}_t^\top \boldsymbol{A} \boldsymbol{c} - \sum_{t=1}^{T} \boldsymbol{\pi}_t^\top \boldsymbol{A} \boldsymbol{c}_t \le \sqrt{\frac{T k_2 \log(k_2)}{2}}.$$

From the above inequalities we can derive that

$$\bar{\boldsymbol{\pi}}^\top \boldsymbol{A} \bar{\boldsymbol{c}} \ge \min_{\boldsymbol{\pi} \in \Pi} \boldsymbol{\pi}^\top \boldsymbol{A} \bar{\boldsymbol{c}} \ge \frac{1}{T} \sum_{t=1}^{T} \boldsymbol{\pi}_t^\top \boldsymbol{A} \boldsymbol{c}_t - \sqrt{\frac{k_1 \log(k_1)}{2T}} \ge \max_{\boldsymbol{c} \in C} \bar{\boldsymbol{\pi}}^\top \boldsymbol{A} \boldsymbol{c} - \sqrt{\frac{k_2 \log(k_2)}{2T}} - \sqrt{\frac{k_1 \log(k_1)}{2T}}.$$

Similarly, we have

$$\bar{\boldsymbol{\pi}}^\top \boldsymbol{A} \bar{\boldsymbol{c}} \le \max_{\boldsymbol{c} \in C} \bar{\boldsymbol{\pi}}^\top \boldsymbol{A} \boldsymbol{c} \le \frac{1}{T} \sum_{t=1}^{T} \boldsymbol{\pi}_t^\top \boldsymbol{A} \boldsymbol{c}_t + \sqrt{\frac{k_2 \log(k_2)}{2T}} \le \min_{\boldsymbol{\pi} \in \Pi} \boldsymbol{\pi}^\top \boldsymbol{A} \bar{\boldsymbol{c}} + \sqrt{\frac{k_1 \log(k_1)}{2T}} + \sqrt{\frac{k_2 \log(k_2)}{2T}}.$$

Table 2: Size of the Nash Support of Games

| Game | Total Strategies | Size of Nash support |
|---|---|---|
| 3-Move Parity Game 2 | 160 | 1 |
| 5,4-Blotto | 56 | 6 |
| AlphaStar | 888 | 3 |
| Connect Four | 1470 | 23 |
| Disc Game | 1000 | 27 |
| Elo game + noise=0.1 | 1000 | 6 |
| Elo game | 1000 | 1 |
| Go (boardsize=3,komi=6.5) | 1933 | 13 |
| Misere (game=tic tac toe) | 926 | 1 |
| Normal Bernoulli game | 1000 | 5 |
| Quoridor (boardsize=3) | 1404 | 1 |
| Random game of skill | 1000 | 5 |
| Tic Tac Toe | 880 | 1 |
| Transitive game | 1000 | 1 |
| Triangular game | 1000 | 1 |

Thus, with $\epsilon_T = \sqrt{\frac{k_1 \log(k_1)}{2T}} + \sqrt{\frac{k_2 \log(k_2)}{2T}}$ we have

$$\max_{\boldsymbol{c} \in C} \sum_{t=1}^{T} \bar{\boldsymbol{\pi}}^\top \boldsymbol{A} \boldsymbol{c} - \epsilon_T \leq \bar{\boldsymbol{\pi}}^\top \boldsymbol{A} \bar{\boldsymbol{c}} \leq \min_{\boldsymbol{\pi} \in \Pi} \boldsymbol{\pi}^\top \boldsymbol{A} \bar{\boldsymbol{c}} + \epsilon_T.$$

By definition, $(\bar{\boldsymbol{\pi}}, \bar{\boldsymbol{c}})$ is $\epsilon_T$-Nash equilibrium.

In situation where both players follow OSO with Less-Frequent Best Response, following Theorem 6 in Appendix B.1 we have:

$$\sum_{t=1}^{T} \boldsymbol{\pi}_t^\top \boldsymbol{A} \boldsymbol{c}_t - \min_{\boldsymbol{\pi} \in \Pi} \sum_{t=1}^{T} \boldsymbol{\pi}^\top \boldsymbol{A} \boldsymbol{c}_t \leq \sqrt{\frac{T k_1 \log(k_1)}{2}} + \sum_{i=1}^{k_1} \alpha_{|T_i|}^i \leq \sqrt{\frac{T k_1 \log(k_1)}{2}} + \sqrt{k1}\sqrt{T}$$

$$\max_{\boldsymbol{c} \in C} \sum_{t=1}^{T} \boldsymbol{\pi}_t^\top \boldsymbol{A} \boldsymbol{c} - \sum_{t=1}^{T} \boldsymbol{\pi}_t^\top \boldsymbol{A} \boldsymbol{c}_t \leq \sqrt{\frac{T k_2 \log(k_2)}{2}} + \sum_{i=1}^{k_2} \alpha_{|T_i|}^i \leq \sqrt{\frac{T k_2 \log(k_2)}{2}} + \sqrt{k_2}\sqrt{T}.$$

Thus, using the same above arguments in the case of OSO, with $\epsilon_T = \sqrt{\frac{k_1 \log(k_1)}{2T}} + \sqrt{\frac{k_2 \log(k_2)}{2T}} + \frac{\sqrt{k_1} + \sqrt{k_2}}{\sqrt{T}}$, we have:

$$\max_{\boldsymbol{c} \in C} \sum_{t=1}^{T} \bar{\boldsymbol{\pi}}^\top \boldsymbol{A} \boldsymbol{c} - \epsilon_T \leq \bar{\boldsymbol{\pi}}^\top \boldsymbol{A} \bar{\boldsymbol{c}} \leq \min_{\boldsymbol{\pi} \in \Pi} \boldsymbol{\pi}^\top \boldsymbol{A} \bar{\boldsymbol{c}} + \epsilon_T.$$

$\square$

# D   Additional Experimental Results

We provide further experiments to demonstrate the performance of the Online Single Oracle and Online Double Oracle algorithms. **All the codes for the experiments are attached as supplementary.**

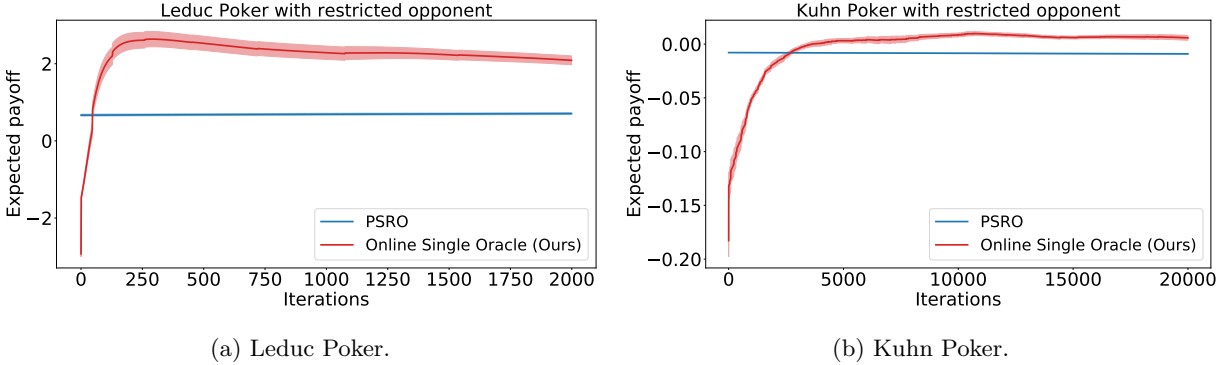

(a) Leduc Poker.

(b) Kuhn Poker.

Figure 5: Performance comparisons against imperfect opponent on Poker games.

**Learning Rate of MWU.** When facing a MWU adversary, we set the learning rate of the adversary to be optimal (i.e., $\mu_t = \sqrt{8 \log(A)/T}$). That is, we eliminate the case when one can exploit MWU adversary with wrong learning rate. For DO-MWU algorithm, we allow the algorithm to update to next subgame when it achieves $\epsilon$-Nash Equilibrium with $\epsilon = 0.01$. Therefore, we optimally choose the learning rate of MWU in a subgame to be $\mu_t = 0.01$ so that the DO-MWU can achieve 0.01-Nash Equilibrium in subgames in at most $O(\log(k)/\epsilon^2)$ steps. For a fair comparison with DO methods, we also choose the same learning rate $\mu_t = 0.01$ for our ODO in the experiments.

**Small Support Size of NEs in Real-World Zero-Sum Games.** Along with the theoretical guarantee in Lemma 1, we demonstrate that small support size assumption holds true for many real-world games. To do that, we report the total strategies (i.e., number of pure strategy for each player) and the size of Nash support for 15 popular real-world games (Czarnecki et al., 2020) in Table 2. As we can see in Table 2, in all of the game we tested, the support size of NEs is a fraction of the game size, reassuring the accuracy of small support size assumption. Notably, in the game with pure NEs (i.e., size of Nash support equals to 1), ODO outperforms other baselines by a large margin, both in exploitability and average payoff.

**Can OSO Exploit an *Imperfect* Opponent?** Finally, we want to test the no-regret property of OSO when the opponent is *imperfect* and it plays a restricted set. We created such an opponent in both Pokers by restricting its strategy set to only 20 pure strategies, and then let it play MWU. We apply our OSO with $\epsilon$-best response as the row player, and compare its perform against PSRO. We run each setting for 10 random seeds. As we can see from Figure 5, OSO quickly achieves positive expected payoff and outperforms PSRO, which is expected because OSO can actively exploit its opponent while PSRO behaves conservatively by playing NE and not exploiting (thus achieving almost constant payoff). Notably, the average expected payoff of OSO decreases slightly in later iterations, we believe it is because the opponent is following MWU, which is also a no-regret method, and both players will converge towards an NE of such a restricted game.

