# OpenReview forum: "Online Double Oracle"
_TMLR — Accepted by TMLR_

### Review · Reviewer_CEob · 2022-08-07

**Summary Of Contributions:**

The paper considers two-player zero-sum normal-form games. The authors propose an algorithm (ODO) that combines the Double Oracle (DO) and the Multiplicative Weight Update (MWU) algorithm. The resulting algorithm keeps the no-regret flavor of MWU, and converges to Nash equilibrium with a rate that depends on the size of the effective strategy set (like DO). The regret bounds of ODO are proven for a time averaged strategy pair.

Empirical work show that the size of the effective strategy set is growing approximately linearly with the the NE support size
in random matrix games, which indicate that ODO could be effective even if the size of the game is large. The algorithm is compared favorably against several baselines on a large number of real (toy) games, including Leduc and Kuhn poker.


**Requested Changes:**

Proof of Theorem 3, equation (4): it is not clear to which Theorem/Corollary in Freund & Schapire (1999) the authors refer, when stating the the regret bound of MWU. In Corollary 4 (FS99), the regret bound is \sqrt{2ln(n)/T} + ln(n)/T. It is possible to obtain \sqrt{ln(n)/(2T))}, but the result should be cited and used precisely.

Step in (6b) should be expanded at least in the appendix, with a clear indication of exactly which inequalities being used.

It should be more explicit in the discussion of the bounds that only the time-averaged strategies converge to a Nash equilibrium, and not the strategies played at any given time. In fact, the bound allows a situation in which at no time the strategies would be even close to a NE (like in a rock-paper-scissors scenario, where each player is playing a pure strategy that beats the opponent's previous action).

In the comparison between bounds, O(\sqrt{log(n)/T}) is regraded as unfavorable to O(\sqrt{k log(k)/T}), when k<<n. But, they are similar when k=log(n), which still can be regarded as k much smaller than n.

In the experiments, Kuhn and Leduc Poker are described as "very large games", but these are toy games that are easy to 'solve' by traditional means. They may have a large number of pure strategy set (it would have been useful to add them in Table 2), but still are fairly small compared to games that are regularly played by humans, or games that have been solved in the last few decades.



Section 3:
	"We consider A_{i,j} to represent the (normalised) loss of the row player when playing a mixed strategy": isn't it pure strategy?

Section 4 title: Oralce -> Oracle

page 9, proof, following "From the above...":
     in "max_{c \in C} \sum ... " the \sum should be deleted

Appendix, Alg. 3, line 5:
	  "arg min arg max" looks a bit strange, even if the intention is clear


**Strengths And Weaknesses:**

The combination of DO and MWU is reasonable, and probably gets the best of both algorithms. I am a bit unsure about restarting MWU each time after a new strategy is added (especially if the effective size is reasonably large), but that is probably useful to simplify the theoretical analysis.

I am not sure if the ideas are scalable for larger games, but that is left for the future, and I think the results of the paper are of interest for researchers working on no-regret algorithms for normal-form games.

Convergence results being provided only for time-averaged strategies seems a bit weaker result than converging to a NE strategy set. This limitation should be made more clear, and discussed clearly when the guarantees of different algorithms are compared.

There are some lesser issues regarding clarity and soundness (listed below) that should be corrected, but these seem easy to fix.

---

> ### Author Response · Authors · 2022-08-27
> **Response to Reviewer CEob [1/2]**
>
> We thank reviewer HYLZ for your time and effort during the review. Below we elaborate further on the reviewer's concerns:
>
> **Q1**: *I am a bit unsure about restarting MWU each time after a new strategy is added (especially if the effective size is reasonably large), but that is probably useful to simplify the theoretical analysis.*
>
> **A1**: In the first paragraph after the proof of Theorem 3 in page 6, we have explained why we restart MWU after each time window. That is, "since we assume a fully adversarial environment, the historical data that the agent learnt in the previous time window does not provide any advantages over the current time window, thus in order to avoid any exploitation, the agent needs to reset the strategy as stated in Algorithm 1". Therefore, changing the starting strategy will not help to improve the theoretical regret bound guarantee in Theorem 3, which works against a fully adversarial environment. In our experiment, we tried to use different starting strategies in each time window. In particular, we use $\frac{1}{k} a_t +\frac{k-1}{k} \bar{\pi_{t-1}}$ , where k is the size of effective strategy set in the current time window, $\bar{\pi_{t-1}}$ is the average strategy from the previous time window and $a_t$ is the new added best response in the current time window. Intuitively, the new starting strategy will be better than the old one if the NE in the previous subgame can perform well in the current subgame. However, we observed that the new starting strategy does not improve the performance of the algorithm, both in average performance and convergence rate to NE. This further explains the reason for choosing the starting strategy as in Algorithm 1.
>
> **Q2**: *Convergence results being provided only for time-averaged strategies seems a bit weaker result than converging to a NE strategy set. This limitation should be made more clear and discussed clearly when the guarantees of different algorithms are compared.*
>
> **A2**:  We are aware of the line of work for the last iterated convergence to NE (e.g., Optimistic MWU), yet we feel that our paper focuses on a different goal (i.e., large size game) rather than the last iterated convergence and thus we have not mentioned it in the paper. For our baselines, we did not use any algorithms with the last iterated convergence guarantee and in fact, algorithms such as Optimistic MWU still inherit the computational complexity of conventional no-regret algorithms such as MWU, thus it is unable to apply in large-scale games. However, for the completeness of the paper, we have made the reader aware of the last iterated convergence results in the last paragraph of Section 5 in the revised version.

---

> ### Author Response · Authors · 2022-08-27
> **Response to Reviewer CEob [2/2]**
>
> **Q3**: *It is possible to obtain $\sqrt{\log(n)/(2T)}$, but the result should be cited and used precisely.*
>
> **A3**: in the revised version, we have cited Theorem 2.2 in Cesa-Bianchi and Lugosi (2006)) for the optimal regret bound $\sqrt{\log(n)/(2T)}$ of MWU (Equation (4)).
>
> **Q4**: *Step in (6b) should be expanded at least in the appendix, with a clear indication of exactly which inequalities being used*
>
> **A4**: We have added further explanation into step (6b) in the revised version.
>
> **Q5**: *It should be more explicit in the discussion of the bounds that only the time-averaged strategies converge to a Nash equilibrium*
>
> **A5**: As explained in **A2**, for the completeness of the paper, we have added more discussion about the last iterated convergence in the revised version (last paragraph of Section 5).
>
> **Q6**: *...they are similar when k=log(n), which still can be regarded as k much smaller than n*
>
> **A6**: when compares with MWU, both computation and regret bound guarantee should be considered. As we have added in Appendix C.4, ODO is more computationally efficient than MWU when $k \leq \sqrt{n},$ which is much larger than the $\log(n)$. We agree that in order to achieve a better regret bound, OSO requires a small k (which is true in many games as shown in Table 2 in the Appendix) yet OSO overcomes the computation infeasibility problem that prevents the application of experts' algorithms such as MWU in large size game (For more discussion about this problem, see e.g., *Online Markov Decision Processes* by Even-Dar et al. (2009), paragraph 3 in Introduction).
>
> **Q7**: *In the experiments, Kuhn and Leduc Poker are described as "very large games", but these are toy games that are easy to 'solve' by traditional means.*
>
> **A7**: As the reviewer pointed out, we call Kuhn and Leduc Poker as "very large games" to emphasis the number of pure strategies they represent (i.e., $2^{12}$ and $3^{936}$ pure strategies). These games will surrender any traditional no-regret algorithms in normal-form games and thus we can show the advantage of our algorithm. Note that in Figure 2, our ODO algorithm (in the normal-form game) can perform not worse than CFR algorithm, an efficient algorithm in solving large extensive form games. Thus, given the result of Kuhn and Leduc Poker experiment, we are optimistic that our algorithms as well as the idea of combining DO and no-regret algorithms can be applied to achieve better performance in much larger games. We leave this important extension as future work.
>
> **Other minor comments**: We thank the reviewer for their thorough review to spot minor issues such as 'mixed strategy', 'Oralce', 'sum'. We have implemented these changes in our revised version.

---

### Review · Reviewer_HYLZ · 2022-08-13

**Summary Of Contributions:**

The paper proposes Online Double Oracle, which combines the ideas of no-regret online learning (e.g., multiplicative weight update) and Double Oracle (for fast computation of NE with small NE support), thus achieving the best of both worlds: no-regret in the adversarial setting and fast convergence to NE in the self-play setting when NE support is small. Experiments on synthetic and real games are provides, showing the advantages of the proposed algorithms over previous ones.

**Broader Impact Concerns:**

There is no concern on ethical implications.

**Requested Changes:**

It would be good to include a discussion on when ODO can find NE in a computationally more efficiently way than MWU+MWU. Below is my discussion that could be added to the paper (but not necessary):

To find a $\epsilon$ NE using ODO, we need $\sqrt{\frac{k\log k}{T}}\leq \epsilon$, where $T$ is the number of rounds. That is, $T\gtrsim \frac{k\log k}{\epsilon^2}$. Since each round requires $O(k)$ computation, the overall computation is roughly $\frac{k^2\log k}{\epsilon^2}$.

To find a $\epsilon$ NE using MWU, we need $\sqrt{\frac{\log n}{T}}\leq \epsilon$. That is, $T\gtrsim \frac{\log n}{\epsilon^2}$. Since each round requires $O(n)$ computation, the overall computation is roughly $\frac{n\log n}{\epsilon^2}$.

Therefore, ODO is more computationally efficient than MWU when $k\lesssim \sqrt{n}$.
(Of course it would be great to provide conditions under which this inequality will hold, or even conditions under which the support of NE is less than $\sqrt{n}$)

**Strengths And Weaknesses:**

Strength
- The proposed idea is simple (in terms of both algorithms and analysis) yet effective.
- Experiments are extensive and demonstrates the benefits of the proposed algorithm.

Weakness
- The theoretical advantage of the proposed algorithm over MWU is unclear. The main reason is that k (the effective support used by the algorithm) may not scale down with the support of the NE if the initial strategy set is not carefully chosen. The authors did not propose solutions (even a heuristic one) to try to make k closer to the true support size.
- While Lemma 3 provides a negative example for a particular initial strategy set (i.e., start with A_{1,1}), it would be desirable to investigate a more reasonable algorithm in which the initial strategy is uniformly chosen from all pure strategies.

Overall, while the idea proposed by the paper is elegant and effective, I feel that some deeper questions (e.g., those stated above) could be further discussed or investigated.

---

> ### Author Response · Authors · 2022-08-27
> **Response to Reviewer HYLZ**
>
> We thank reviewer HYLZ for your time and effort during the review. Below we elaborate further on the reviewer's concerns:
>
> **Q1**: *theoretical advantage of the proposed algorithm over MWU is unclear...*
>
> **A1**: while in general, we can not theoretically bound the size of the effective strategy set (i.e., $k$) with the Nash's support size as shown by a counter-example in Lemma 3, practically we show that in many games (Figure 1), there is a linear relationship between these quantities and therefore our algorithm will be more efficient than MWU in these cases. Furthermore, in Figures 3 and 4, we show that ODO outperforms MWU in many games, both in convergence to NE and in average payoff, reassuring that ODO is better than MWU in many practical games. Given the hardness result in Lemma 3 for the general setting, we agree with the reviewer that finding a large subclass of game in which we can theoretically bound the size of effective strategy and the Nash's support size will be a breakthrough in DO/PSRO line of works, yet despite many papers show the success of DO/PSRO in practice, there is no success in providing this theoretical guarantee. Our paper does not try to overcome this difficulty in DO/PSRO method (even though in Lemma 2, we provide a small subset of game with dominant strategy so that we can theoretically bound $k$), but rather shows that the practical success of DO/PSRO method can also be used in online learning algorithms to achieve better performance. We leave the valid concern over the exact relationship between $k$ and NE's support as an important and breakthrough future work.
>
> **Q2**: *While Lemma 3 provides a negative example for a particular initial strategy set (i.e., start with $A_{1,1}$), it would be desirable to investigate a more reasonable algorithm in which the initial strategy is uniformly chosen from all pure strategies.*
>
> **A2**: The reviewer raised a valid point to have an initial strategy as uniformly chosen from all pure strategies. Fortunately, the example in Lemma 3 can also work in this case. Suppose that the algorithm starts with the pure strategy $i$ (and thus the initial subgame will be $A_{i, i}$). Then following the exact same argument as in Lemma 3, we will derive that it will take $n-i+1$ subgames ($k=n-i+1$) before the algorithm reaches the NE. Intuitively, as the game in Lemma 3 is designed such as the NE of the subgames will move from the top to the bottom of the diagonal, then changing the starting subgame does not break the overall process. Therefore, if we choose the initial strategy as uniformly random, then the expected size of the effective strategy will be: $\frac{\sum_{i=1}^n n-i+1}{n}=\frac{n+1}{2}$. Therefore, the hardness result in Lemma 3 is still valid when we choose the starting point as random. We thank the reviewer for the suggestion and have added the argument in our revised version (below Lemma 3).
>
> **Q3**: *ODO is more computationally efficient than MWU when
> $k \leq \sqrt{n}$. (Of course it would be great to provide conditions under which this inequality will hold, or even conditions under which the support of NE is less than $\sqrt{n}$)*
>
> **A3**: As we answered in **Q1**, theoretically connecting $k$ and Nash support is impossible in general, following the hardness results in Lemma 3. Therefore, we consider subclasses of games such as games with strictly dominant strategy (Lemma 2), asymmetric games (Lemma 1) as well as using experimental results (Figure 1 and Table 2) to demonstrate that indeed small $k$ or small support of NE covers a large set of games (e.g., $k \leq \sqrt{n}$). With the current theoretical and practical results in the paper, it is safe to say that ODO is more efficient than MWU in both computation and performance in most practical games considered. We thank the reviewer for the computational analysis of ODO and MWU. We will add them in along with the result of Lemma 1 and 2 in the revised version (below Lemma 2 and Appendix C.4) so that we have two subclass of game such as ODO theoretically has better computation and regret bound than MWU (e.g., in Lemma 2, we also require $m \leq \sqrt{n}$).

---

### Review · Reviewer_tPpz · 2022-08-15

**Summary Of Contributions:**

This paper proposes a new algorithm Online Double Oracle (ODO) for finding Nash Equilibria (NE) in two-player zero-sum normal-form games with a large number of actions.

On the theoretical side, the paper shows that ODO achieves both convergence to NE and handles large action spaces like Double Oracle (DO) type algorithms, as well as no-regret against adversarial opponents when deployed by a single agent as Online Single Oracle (OSO). The regret can be better than standard Multiplicative Weights Update (MWU) if the learned policy at termination has a small support size.

On the empirical side, the paper shows that ODO often outperforms standard DO when used as self-play to learn NE, as well as better payoff than MWU when played against an (adversarial) opponent who runs their own MWU algorithm.


**Requested Changes:**

Please find my questions in the Weaknesses part above.

Additional questions / minor suggestions about writing and discussions:

* Page 1, “Firstly” and “Secondly”, in my opinion they are really just one thing and “Secondly” is the main thing. “Firstly” was talking about required coordination to learn NE, but in a weaker sense, coordination is almost always required, e.g. with self-play + no-regret type algorithms, at least the two players each need to deploy a no-regret algorithm. It’s only that DO seems to be a stronger coordination requirement (very precise algorithm steps) where no-regret could be a weaker requirement (any no-regret algorithm suffices).

* Table 1, “Large Games”, perhaps good to briefly explain what that means in the caption, to make the table self-contained (even though it’s already explained in the main paper).

* Section 3.1 (Page 3), “RL methods are adopted to approximate the best-response strategy”, what does “RL methods” refer to? e.g. perhaps good to add “(e.g. those with neural net function approximation)” if that’s what the authors meant?

* Section 4.2 (Page 6), “Generally”, the argument about effective strategy set sounds a bit hand-wavy to me; in a strict sense it lacks considerations about sensitivity (e.g. if the opponent only plays $\epsilon$-NE and we are learning their $\epsilon$-best response, then we may still end up getting a policy with large support. If the paragraph was meant to be a heuristic argument, it may be good to be explicit about that (e.g. say “heuristically, …”).

* Lemma 2 (Page 7), what is a “strictly dominant strategy”? Maybe add a definition in Appendix C.2 and then add a pointer here?

* Theorem 5 (Page 9), may be good to shorten or put to appendix, and mention it is just the standard online-to-batch argument for converting regret to Nash.

* Section 5.1, “every time a sub-game NE is solved, which often requires thousands of MWU iterations”... sounds too hand-wavy? For example, why can’t I carefully chose the precision parameter in each iteration of DO-MWU in a decaying fashion, so that overall number of MWU iterations are controlled yet DO-MWU algorithm still performs well? (cf. the same question in the “weakness” part about experiments).

* Section 5.1, “even if DO implements MWU to solve the sub-game NE, it is still not a no-regret algorithm”, why is that (good to add a reference?)

* Section 6.2, “exploitability (i.e., the distance to a true NE)”, the description could be misleading (e.g. regarded as parameter-space distance). Consider changing to a more rigorous description of exploitability?


**Strengths And Weaknesses:**

Strengths:

* The proposed ODO algorithm combines advantages from two disparate lines of algorithms for learning NE: DO type algorithms (which grows support size incrementally and thus handles large action space), and no-regret algorithms (who performs well against adversarial opponents when deployed by a single agent). Notably, it is a standard result that MWU can be used to learn NE too via self-play, but MWU cannot handle large action spaces. The very promise of DO is to get around that by growing support size one at a time, but DO loses no-regret when deployed by a single agent. In this sense, ODO gives sort of a best-of-both-worlds. As the core claim of the paper, this is a nice contribution in my opinion.

* Theorem 3 shows that the regret bound of OSO scales only with $k$, the support size of the final output policy. The proof of this fact appears not to be technically challenging, and follows directly from summing the standard regret bounds for MWU over the $k$ periods. But logically, it gives a justification of why ODO should be used when regret is the main consideration (i.e. why not just use MWU in that case).

* I like the experiments section a lot. First of all I think the *designs* of the three experiments in Sections 6.1-6.3 are already nice and accurately test the main claims of the paper, in particular Section 6.1 which tests the support size of the final output policy when true game has small support. The experiments are well executed and complete enough (though probably not super extensive). The results look promising and reasonably justify the main claims in my opinion.

* Overall the presentation and logical structure of the paper is nice too. The related works and various discussions are quite clear. (e.g. I like having Section 5.1 which compares ODO with DO+MWU; I had exactly that question in mind before reaching there :) I also like Lemma 3 which gives a negative result about the support size of ODO’s final output policy in the worst case.


Weaknesses:

* A main concern is that the choice of the stepsize in MWU algorithms / subroutines seem to be unspecified. If they are indeed not properly chosen, this could affect many of the main theoretical and empirical claims:
  - The stepsize of MWU is unspecified in both the ODO algorithm box (Algorithm 1) and Theorem 3, whose proof uses the standard fact that MWU with $A$ actions achieves $\sqrt{\log A\cdot T}$ regret. However, this requires careful choice of the stepsize: (1) Either $T$ is known in advance and the stepsize is chosen as $\sqrt{\log A/T}$, or (2) if an anytime algorithm (i.e. with changing learning rate) is used. The former cannot hold for ODO algorithm because the algorithm could not know the length of the current period in advance. The later is a potential remedy, e.g. by using the anytime FTRL algorithm (cf. Exercise 28.13 in Szepesvari and Lattimore, “Bandit Algorithms”), but that would substantially change the algorithm (FTRL with changing learning rate is no longer equivalent to MWU).

  - In the experiments, I did not spot how the stepsizes for MWU are chosen. This importantly affects conclusions about both ODO (which uses MWU as subroutine), as well as other baselines such as DO+MWU, and an opponent who plays an MWU. In particular, this makes the claim in Section 6.3 that ODO > MWU slightly fishy (e.g. an MWU algorithm with the correct learning rate already beats an opponent who runs MWU with the wrong learning rate, e.g. too small).

  - A related question about DO-MWU: how many MWU iterations are performed in each DO loop? I think being careful here, e.g. by choosing different #iterations per loop, may substantially improve the DO-MWU baseline?

* Aren’t Lemma 1 and 2 or results of similar nature known in the literature? For example, Lemma 1 shows that for an $n\times m$ game with $n\ge m$ and a unique NE, the support size of the NE for both players is bounded by $m$. However, isn’t this just a corollary of the fact that all NE of nondegenerate $n\times m$ games has equal support sizes for both players (Proposition 3.3, “Algorithmic Game Theory”, Nisan et al. 2007)? If that’s the case, perhaps proper citations around these Lemmas would be deserved.


* The empirical result that the final output policy has a small support size when true game has a sparse NE in Section 6.1, is claimed as an advantage of ODO. How does this compare with the Regret Matching algorithm, which also has some “pruning” effect and maintains a sparse strategy in later iterations, if there are only so many good actions? (Though, I understand Regret Matching does not handle large action spaces as nicely as ODO.)

Given the above, I am positive about the core contribution and overall execution of the paper, provided the above concerns (especially the one about stepsize) can be addressed.

---

> ### Author Response · Authors · 2022-08-27
> **Response to reviewer tPpz [1/3]**
>
> We thank reviewer tPpz for your time and effort during the review. Below we highlight the change we made in the paper as well as elaborate further on the reviewer's questions:
>
> **Q1**: *choice of the stepsize in MWU algorithms / subroutines seem to be unspecified*
>
> **A1**: The reviewer raises a valid point of choosing the correct stepsize for our subroutine MWU. As the reviewer argued, if the agent does not know the time window $T_i$, he can not simply choose the fixed optimal stepsize $\sqrt{8log(A)/T_i}$ in this time window. In order to define the choice of stepsize rigorously, we add further explanation to the revised version (Definition 2, Appendix A.2 and A.3). For an unknown $T$, firstly, we can naively apply the doubling trick (Appendix A.2), which decreases the stepsize in every $2^m$ timestep for $m$ in $[1,2,...,\log_2(T)]$. That way, the regret in each time window will be bounded by $\frac{\sqrt{2}}{\sqrt{2}-1}\sqrt{log(A)/2T}$ and therefore the regret of OSO in Theorem 3 will be: $\frac{\sqrt{2}}{\sqrt{2}-1} \frac{k \log(k)}{\sqrt{2T}}$, which is worse than the original one by a constant factor. Secondly, we can use the adaptive learning rate version of MWU. That is, we use AdaHedge algorithm in *follow the leader if you can, hedge if you must* by Rooij et .al(JMLR 2014). Specifically, by adapting the learning rate in MWU by $\mu_t= \frac{\log(A)}{\Delta_{t-1}}$, where $\Delta_t$ denotes the cumulative mixability gap and is nondecreasing in $t$, we can achieve the regret of $\sqrt{log(A)T}+O(1)$ in the worst case scenario (Appendix A.3). Therefore, when $T_i$ is unknown, by using the Doubling Trick or AdaHedge, we can maintain the regret bound as stated in Theorem 3 up to a fixed constant factor. Thus, we can always maintain the regret bound of $O(k \log(k) T)$ for OSO both when $T_i$ is known and unknown. For simplicity, we use the optimal regret bound of MWU in our analysis (further explanation can be found on page 4 of the revised version).
>
> **Q2**: *the claim in Section 6.3 that ODO $>$ MWU slightly fishy (e.g. an MWU algorithm with the correct learning rate already beats an opponent who runs MWU with the wrong learning rate, e.g. too small).*
>
> **A2**: In all of our experiments, when we consider MWU as an opponent or as a baseline, we always choose the optimal fixed learning rate $\sqrt{8log(A)/T}$ for the opponent so that we test the situation where we face the best opponent. We have added further explanation in Appendix D of the revised version.
>
> **Q3**: *by choosing different number of iterations per loop, may substantially improve the DO-MWU baseline?*
>
> **A3**: The main idea of DO-MWU is to find the Nash equilibrium in each subgame before updating the subgame. Therefore, the number of MWU iterations in each subgame can not be chosen in advance, but rather depend on the rate of convergence to NE in each subgame. In the experiment of DO-MWU, we allow DO-MWU to update the next subgame when it achieves $\epsilon$-Nash Equilibrium with $\epsilon=0.01$ (Further reduction in $\epsilon$ will lead to a better approximation of NE but require a larger number of iterations). For the learning rate, since the aim of the algorithm is to achieve the $\epsilon$-Nash Equilibrium, which can be done by achieving $\epsilon$ average regret, we pick the learning rate of MWU in DO-MWU subgame to be $\mu=0.01$ so that the algorithm can achieve $0.01$-Nash Equilibrium in at most $O(\log(N)/\epsilon^2)$ steps. Note that we will stop the MWU in DO-MWU when it achieves $\epsilon$-Nash Equilibrium, not until the maximum iteration is reached. For a fair comparison with DO methods, we also choose this learning rate for our ODO in the experiments. We have added these detailed experimental setups in Appendix D of the revised version.

---

> ### Author Response · Authors · 2022-08-27
> **Response to reviewer tPpz [2/3]**
>
> **Q4**: *Lemma 1 shows that for an $n\times m$ game with  $n\geq m$ and a unique NE, the support size of the NE for both players is bounded by $m$ .However, isn’t this just a corollary of the fact that all NE of nondegenerate  games has equal support sizes for both players (Proposition 3.3, “Algorithmic Game Theory”, Nisan et al. 2007)*
>
> **A4**: Lemma 1 is not a corollary of Proposition 3.3 in Algorithmic Game Theory. The reason is, a game with a unique Nash equilibrium can be a degenerate one and thus Proposition 3.3 will not hold. For example, take the zero-sum game with the following game matrix:
>
> $$\left[\begin{array}{ccc}
> 2 & 1 & 1\\\\
> 3 & 1 & 1 \\\\
> 3 & 1 & 1\\\\
> \end{array}\right]$$
> In this game, there exists a unique NE: $[(1,0,0),(1,0,0)]$. Yet, this game is degenerate since when the column player follows the mixed strategy (0,0.5, 0.5) with support of $2$, there are $3$ best responses for the row player.
>
> However, Proposition 3.3 covers many important samples that our Lemma 1 does not consider, namely games with no unique equilibrium but non-degenerate. Therefore, along with Lemma 1, we have mentioned and cited Proposition 3.3 in the revised version (after Lemma 1) to further validate our assumption about the small support size of the NE.
>
> **Q5**: *How does this compare with the Regret Matching algorithm, which also has some “pruning” effect and maintains a sparse strategy in later iterations, if there are only so many good actions?*
>
> **A5**: Regret Matching will be computationally inefficient in large size games since it still requires maintaining the cumulative regret for each pure strategy, thus its computation in each round will depend on the size of the game. In our experiments, instead of comparing with Regret Matching directly, we compare ODO with CFR, an efficient algorithm in extensive form games that uses the idea of regret matching in each information set. Since ODO can perform well against CFR (Figure 2), we believe that it will outperform the naive Regret Matching in large-size games (when it does not consider information sets as in CFR). Note here that the idea we introduce in this paper can be applied to any no-regret algorithms in normal-form games, thus it is interesting to see which online learning algorithm can return the best performance in practical when applying our technique (including Regret Matching). We leave it as an important future work.

---

> ### Author Response · Authors · 2022-08-27
> **Response to reviewer tPpz [3/3]**
>
> Minor suggestions about writing and discussions:
>
> **Q6**: *“Firstly” and “Secondly”, in my opinion, they are really just one thing and “Secondly” is the main thing.*
>
> **A6**: As highlighted in the paper, we agree that the "secondly" argument is the more important one. Yet, the first point does not only imply that both agents need to follow no-regret algorithms with precise step, they also require that the agents only choose the actions in the subgame. Therefore, without the coordination to know the action in each subgame, the DO method will fail to apply.
>
> **Q7**: *“Large Games”, perhaps good to briefly explain what that means in the caption*
>
> **A7**: We have included the explanation in the table.
>
> **Q8**: *what does “RL methods” refer to?*
>
> **A8**: Many RL algorithms can be adapted to approximate the best-response strategy. For clarification, we include Soft Actor Critic as an example.
>
> **Q9**: *“Generally”, the argument about effective strategy set sounds a bit hand-wavy to me*
>
> **A9**: We agree with the reviewer on this. "Heuristically" is more appropriate in this case.
>
> **Q10**: *"what is a “strictly dominant strategy”?"*
>
> **A10**: we have included the definition in the Appendix C.2.
>
> **Q11**: *why can’t I carefully chose the precision parameter in each iteration of DO-MWU in a decaying fashion, so that overall number of MWU iterations are controlled yet DO-MWU algorithm still performs well?*
>
> **A11**: As we have explained in **Q3**, in our experiment, we have chosen the optimal learning rate for DO-MWU to achieve $\epsilon$-NE, yet it still requires thousands of rounds before converging. The reason is that changing the learning rate can not result in a better regret bound than $O(\sqrt{\log(A) T})$, thus theoretically, the number of iterations that MWU required to converge will be proportional to $1/\epsilon^2$. Since DO method will require the $\epsilon$ to be small, it will take thousands of MWU iterations in each subgame.
>
>
> **Q12**: *“even if DO implements MWU to solve the sub-game NE, it is still not a no-regret algorithm”, why is that (good to add a reference?)*
>
> **A12**: Final output of DO-MWU is a NE of the game, yet following the NE is not a no-regret algorithm in the general case (i.e., there is a better-fixed strategy against a specific opponent).
>
> **Q13**: *“exploitability (i.e., the distance to a true NE)”, the description could be misleading (e.g. regarded as parameter-space distance)*
>
> **A13**: To be precise, we have changed that to (exploitability$=0$ implies the algorithm reaches the true NE).

---

> ### Comment · Reviewer_tPpz · 2022-09-23
> **After response**
>
> I thank the authors for the very detailed response and paper revision. I think most of my concerns (learning rates, experimental details, writings) are addressed, and the revised paper is much more complete in my opinion.

---

### Author Response · Authors · 2022-08-27
**General Response**

We thank 3 reviewers for their thorough review of our paper. Overall we feel that all reviewers are positive about the current contributions of the paper on online learning and DO methods literature. Furthermore, many important future works can be developed based on our paper (e.g., a deeper theoretical understanding of the relation between ODO/DO) iterations vs NE's support and applications to more practical games). About the concerns that the reviewers raised over the paper, we think that all of them are addressable and we have implemented changes in our revised version. Due to the time constraint for the response, if we have not addressed any concerns properly, please let us know so we can make it happen in our next version. Below, we respond to comments that each reviewer makes during the review.

---

### Decision · Action_Editors · 2022-09-25

**Recommendation:** Accept as is

**Comment:**

I thank the authors for a very diligent rebuttal - all the reviewers were satisfied with it and therefore I suggest to accept ODO as is.

---

> ### Author Response · Authors · 2022-10-03
> **Camera-ready version**
>
> We thank the Action Editor and all the reviewers for their constructive comments. We have uploaded the camera-ready version as required. Again, we really appreciate all the hard works that everyone has put into improving the paper.